# Chromosome compartments on the inactive X guide TAD formation independently of transcription during X-reactivation

Moritz Bauer [1], Enrique Vidal[1], Eduard Zorita [1], Nil Üresin[1], Stefan F. Pinter [2], Guillaume J. Filion [1,3,4,5 ✉] & Bernhard Payer [1,3,5 ✉]

A hallmark of chromosome organization is the partition into transcriptionally active A and repressed B compartments, and into topologically associating domains (TADs). Both structures were regarded to be absent from the inactive mouse X chromosome, but to be re-established with transcriptional reactivation and chromatin opening during X-reactivation. Here, we combine a tailor-made mouse iPSC reprogramming system and high-resolution Hi-C to produce a time course combining gene reactivation, chromatin opening and chromosome topology during X-reactivation. Contrary to previous observations, we observe A/B-like compartments on the inactive X harbouring multiple subcompartments. While partial X-reactivation initiates within a compartment rich in X-inactivation escapees, it then occurs rapidly along the chromosome, concomitant with downregulation of *Xist*. Importantly, we find that TAD formation precedes transcription and initiates from Xist-poor compartments. Here, we show that TAD formation and transcriptional reactivation are causally independent during X-reactivation while establishing Xist as a common denominator.

[1] Centre for Genomic Regulation (CRG), The Barcelona Institute of Science and Technology, Dr. Aiguader 88, Barcelona, Spain. [2] Department of Genetics and Genome Sciences, Institute for Systems Genomics, University of Connecticut Health Center, Farmington, CT, USA. [3] Universitat Pompeu Fabra (UPF), Barcelona, Spain. [4] Present address: Dept. Biological Sciences, University of Toronto Scarborough, Toronto, ON, Canada. [5] These authors jointly supervised this work: Guillaume J. Filion, Bernhard Payer. ✉email: guillaume.filion@gmail.com; bernhard.payer@crg.eu

To achieve gene dosage balance between males (XY) and females (XX), mammals transcriptionally inactivate one of the two X chromosomes in females during early embryonic development in a process called X-chromosome inactivation (XCI). While the active X chromosome resembles in many aspects an autosome, the inactive X (Xi) has a unique repressive configuration and chromosome conformation, which sets it apart from other chromosomes. This has established XCI as a unique model to study the formation of heterochromatin and the mechanisms of chromosome folding and chromosome organization[1–3]. Mammalian chromosomes have been shown to be organized in two separate compartments: A, corresponding to open chromatin and high RNA expression, and B, corresponding to closed chromatin and low expression[4]. Moreover, on a more fine-scaled level, chromosomes have been shown to be partitioned into megabase-sized local chromatin interaction domains, termed topologically associating domains (TADs), whose boundaries are enriched for the insulator binding protein CTCF and cohesin[5,6]. Intriguingly, both these levels of chromatin organization appear to be absent or attenuated on the inactive mouse X chromosome. Spatial proximity maps obtained by Hi-C from neural precursors cells (NPCs)[7,8], as well as from fibroblasts[9,10], have suggested that the mouse Xi lacks compartments, with exception of weak compartmentalization observed in mouse brain cells[10]. Furthermore, the Xi was shown to display a global attenuation of TADs[8], apart from regions escaping X-inactivation[7,11,12]. While the mouse Xi has therefore been considered to be mostly "unstructured", an exception is its unique bipartite organization in two so-called mega-domains that are separated by a tandem repeat locus, Dxz4[7,10,13,14]. The key player in the formation of the silenced X chromosome and ultimately its unique chromosome conformation is the long non-coding RNA Xist. Xist coats the X from which it is expressed and silences the chromosome through the combined action of multiple interaction partners that set up a heterochromatic environment[11,15,16]. During this process, Xist repels architectural proteins like CTCF and Cohesin[11], thereby actively contributing to the attenuation of TADs[8] and leading to the distinct chromosome conformation[7,17,18] of the Xi. There has been intense research effort to understand the dynamics of transcriptional silencing, the mechanisms of transition to the unique structure of the Xi and the connection between the two processes[8,9,19,20], but how the process is reversed during the reactivation of the X chromosome has received attention only recently[1,21,22].

In mice, X-reactivation occurs twice during early development. The first round takes place at the blastocyst stage within the pluripotent epiblast of the inner cell mass[23–26]. This allows the female embryo to switch from an imprinted form of X-inactivation, whereby the X inherited from the father's sperm is inactivated, to a random form where either X can be inactivated. The second round of X-reactivation takes place in primordial germ cells during their migration and colonization of the gonads, ensuring that an active X can be transmitted to the next generation[27–30]. However, while mechanistically, both share common features like the downregulation of Xist and the erasure of silencing marks like H3K27me3, their kinetics differ greatly, as X-reactivation in the blastocyst occurs within a day, while it takes several days during germ cell development.

X-reactivation can also be studied in vitro: induced pluripotent stem cells (iPSCs) generated from somatic cells through reprogramming have two active X chromosomes[25,31–34], linking the X-reactivation process to de-differentiation into the naive pluripotent stem cell state[35]. However, the mechanisms that govern the transition have not yet been elucidated. It is unclear how the interplay of sequence, 3D-structure, chromatin status, and trans-acting factors affects the reactivation of X-linked genes and why

X-reactivation in vitro during iPSC reprogramming is a slow and gradual progress as recently proposed[32]. In particular, it is unknown if the dramatic topological rearrangement of the X chromosome from an inactive state with two mega-domains into an autosome-like active state consisting of A/B-compartments and TADs[7,11,13,14] occurs before X-linked genes are reactivated, or rather follows transcription as observed during XCI[36]. This is especially relevant from a general gene-regulatory point of view, as cause and effect between chromosome topology and transcriptional activity have been under debate[34,37,38]. The dramatic transcriptional and structural remodelling of an entire chromosome from an OFF to an ON state makes X-reactivation a particularly attractive model system to address these important questions.

In this work, we investigate the temporal dynamics of transcriptional reactivation, chromatin opening, and reveal their relationship to the topological rearrangement of the inactive X in an optimized iPSC reprogramming system. We show that the inactive X contains A/B-like compartments that guide TAD formation, before the onset of transcriptional reactivation.

## Results

**PaX: a tailor-made reporter model system for X-chromosome reactivation.** Previous studies on X-chromosome reactivation during iPSC reprogramming were based on mouse embryonic fibroblast (MEF) reprogramming[25,31–33] and have been mitigated by several limitations inherent to these systems. First, reprogramming and X-reactivation efficiencies were low so that it has been difficult to study the process by assays that relied on a large number of cells, such as Hi-C. Second, the heterogeneity of the samples was high, with cells of different reprogramming stages and degrees of X-reactivation represented in single populations.

We therefore created an optimized in vitro model system called PaX (Pluripotency and X chromosome reporter) designed to overcome these pitfalls (Fig. 1a). PaX is based on a hybrid female embryonic stem cell (ESC) line[39,40], that contains one Mus musculus (Xmus) and one Mus castaneus (Xcas) X chromosome, allowing us to distinguish the two X based on sequence polymorphisms. This cell line is karyotypically highly stable without detectable X-chromosome loss during prolonged culture[39] (Supplementary Fig. 1a, b and methods), an important prerequisite for X-inactivation and -reactivation studies. Furthermore, the cell line harbours a Tsix truncation (TST) on Xmus, forcing a biased inactivation during differentiation[40,41]. PaX ESCs are differentiated into neural precursor cells (NPCs), to consistently obtain a large and homogeneous population of somatic cells that have undergone X-chromosome inactivation[42]. iPSC reprogramming of NPCs is initiated by the addition of doxycycline which triggers the expression of an optimized all-in-one doxycycline-inducible MKOS (c-Myc, Klf4, Oct4, Sox2) reprogramming cassette[43]. During the reprogramming process, the pluripotency- (P-RFP, driven by a Nanog promoter fragment) and X-reporters (X-GFP)[44] are used to isolate first, pluripotent cells poised for X-reactivation and later, cells having undergone X-reactivation. The unique features of the PaX system enabled us to isolate pure populations of cells at different stages of X-reactivation allowing us to chart a roadmap of the X-reactivation process with high time resolution (Fig. 1a).

First, we assessed the kinetics of reprogramming markers with our PaX reporter cell line. After initiation of reprogramming, our cell line first upregulates the pluripotency marker stage-specific embryonic antigen 1 (SSEA1) with subsequently around 15-25% of SSEA1+ cells becoming P-RFP+ (Supplementary Fig. 1c), of which up to about 85% reactivate X-GFP (Supplementary Fig. 1d). To test if P-RFP+/X-GFP− cells therefore represented a

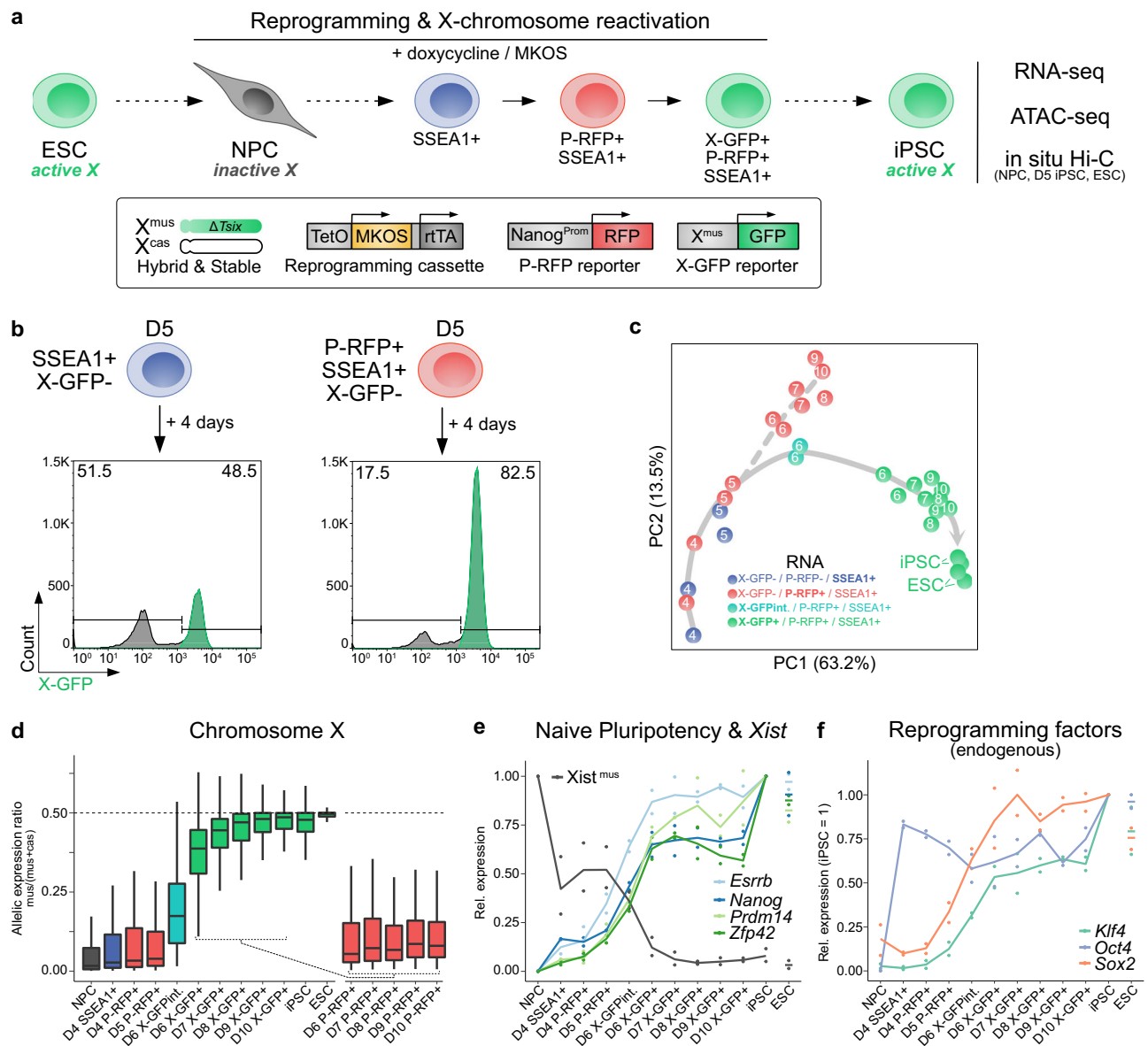

**Fig. 1 A tailor-made reprogramming system to efficiently trace X-chromosome reactivation. a** Schematic representation of the PaX reprogramming system. **b** X-reactivation efficiency of indicated reprogramming intermediates isolated on day 5 and then reprogrammed for an additional 4 days. Shown are representative histograms gated on SSEA1+ cells. Numbers indicate the percentages of X-GFP+ and X-GFP– cells. **c** PCA of gene expression dynamics during reprogramming (n = 12,318 genes). Solid grey arrow, hypothetical trajectory. Dashed grey arrow, samples deviating from main reprogramming trajectory. **d** Allelic expression ratio (mus/(mus+cas)) of protein-coding genes expressed from chromosome X (n = 335). For biallelic expression, ratio = 0.5. Box plots depict the first and third quartiles as the lower and upper bounds of the box, with a band inside the box showing the median value and whiskers representing 1.5x the interquartile range. **e** Average gene expression kinetics of naive pluripotency genes *Esrrb*, *Nanog*, *Prdm14*, and *Zfp42* (relative to the levels in iPSC) and Xist^mus (relative to the levels in NPC) during reprogramming (n = 2). **f** Average endogenous gene expression kinetics of the reprogramming factor genes *Klf4*, *Oct4*, and *Sox2* during reprogramming (n = 2, relative to the levels in iPSC). Endogenous expression assessed via the genes' 3'-UTR.

pluripotent population primed for X-reactivation, we isolated SSEA1+/P-RFP–/X-GFP– and SSEA1+/P-RFP+/X-GFP– cells on day 5 by fluorescence-activated cell sorting (FACS) and continued reprogramming. Analysis 4 days later showed that while only half of the SSEA1+/P-RFP– cells were able to reactivate X-GFP, the number rose to around 80% for SSEA1+/P-RFP+ (Fig. 1b). We conclude that our PaX system enables us to separate homogeneous cell populations, which is a prerequisite for a faithful kinetic analysis of the X-reactivation process.

Utilizing this specialized reprogramming system, we set out to obtain a high-resolution map of X-reactivation in relation to the iPSC reprogramming process. We performed differentiation of PaX ESCs into NPCs followed by reprogramming to iPSCs and FACS sorted reprogramming stages in 24 h intervals, utilizing the PAX reporters to enrich for subpopulations primed for, and later having undergone, X-reactivation. On these subpopulations, we performed allele-specific RNA-seq and ATAC-seq (Assay for Transposase-Accessible Chromatin with High Throughput Sequencing) to reveal gene reactivation and chromatin opening kinetics, respectively. We also performed in situ Hi-C at three key stages (NPCs, D5 and ESCs) to get the first overview of the structural changes during X-reactivation at high resolution.

To define the trajectory towards X-reactivation, we performed principal component analysis (PCA) of the RNA-seq data. As day 4 (D4) samples clustered far away from NPCs (Supplementary Fig. 1e), we repeated this analysis excluding NPCs to improve the resolution along the reprogramming time course (Fig. 1c). This revealed a trajectory of reprogramming and X-reactivation along enriched subpopulations that would have otherwise been merged. Subpopulation of cells which failed to undergo X-reactivation deviated from this trajectory (Fig. 1c), confirming X-reactivation as a hallmark feature of cells on a successful iPSC reprogramming path[32,33,45]. Next, to determine X chromosome-wide gene reactivation kinetics along this trajectory, we assessed the allelic expression ratio between the inactive $X^{mus}$ and the active $X^{cas}$, for 335 genes which had sufficient allelic information and expression (see methods). Whereas D6 P-RFP+/XGFP− cells still portrayed inactivation levels similar to NPCs, we found a clear switch in D6 P-RFP+/X-GFP+ cells, displaying an allelic ratio close to iPSCs (Fig. 1d), showing reactivation of the inactive $X^{mus}$ in this population. In contrast, genes on chromosome 13 maintained a consistent biallelic expression throughout reprogramming (Supplementary Fig. 1f), confirming that these allelic changes were specific to the inactive $X^{mus}$.

In parallel, we observed the characteristic sequential activation of endogenous pluripotency factors (Fig. 1e, f), a key event during iPSC reprogramming[34,46–48]. First, we could detect high endogenous expression levels of *Oct4* on D4, at which point we also observed a sharp drop in expression markers of neural precursor cells *Blbp*, *Nestin*, *Pax6*, and *Sox1*, showing the rapid extinction of the somatic gene expression signature (Supplementary Fig. 1g). On D5, we observed activation of *Sox2* and *Esrrb* and on D6 upregulation of naive pluripotency factors *Klf4*, *Nanog*, *Prdm14*, and *Zfp42/Rex1*, coinciding with downregulation of the X-inactivation master regulator *Xist* (Fig. 1e). This is consistent with *Xist* downregulation during iPSC reprogramming being dependent on binding of both core and naive pluripotency factors along their binding hubs at the X-chromosome inactivation centre (*Xic*)[25,35,49–51]. In conclusion, the unique properties of the PaX system revealed that X-reactivation during iPSC reprogramming occurs rapidly and is tightly linked to the establishment of the naive pluripotency program and the downregulation of *Xist*. Thereby it faithfully mirrors the rapid X-reactivation kinetics observed in mouse blastocysts in vivo[26].

**An underlying A/B-like compartmentalization persists on the inactive X.** Previous studies have shown that the active and inactive X chromosome have strikingly different 3D conformations[6,7,11,13,14,17,52]. In particular, while the active mouse X chromosome has an autosome-like structure, exhibiting active A and inactive B compartments and topologically associating domains (TADs), the inactive X chromosome is thought to lack compartmentalization and to solely exhibit TADs of attenuated strength[8]. Moreover, the inactive X consists of two mega-domains divided by the boundary element *Dxz4* with TAD structures only around genes that escape the X-inactivation process[7]. The so-called "unstructured" state of the inactive X is shaped by Xist RNA, which repels CTCF and cohesins, thereby causing the loss/attenuation of TADs[11]. Furthermore, SMCHD1, was shown to be responsible for the merging of compartments on the inactive X chromosome, ultimately leading to its compartmentless structure[8]. We thus obtained the contact map of the X chromosome using Hi-C to determine how structural remodelling ties in with chromatin opening and transcriptional reactivation during X-reactivation.

We performed in situ Hi-C[14,34] on NPCs to assess the inactive X, on D5 P-RFP+/X-GFP− pre-iPSCs to investigate possible structural changes immediately before the reactivation of gene expression, and on ESCs to capture cells with two active X and as an endpoint, as it has been shown previously that 3D genomes of iPSCs and of ESCs are overall highly similar[53] (Fig. 2a and Supplementary Fig. 2a). Furthermore, we FACS-sorted G1 cells based on DNA content (Supplementary Fig. 2b), which reduces cell cycle-induced variability and recovers a greater proportion of long-range *cis* contacts of samples[54]. Replicates were highly correlated[55] (Supplementary Fig. 2c) and reached a considerably higher resolution than comparable studies assessing the murine inactive X using in situ Hi-C (Supplementary Fig. 2d)[8,9]. Visual inspection of the Hi-C matrices revealed the characteristic presence of the two mega-domains on the inactive $X^{mus}$, not only in NPCs but also in D5 P-RFP+ cells (Fig. 2a), showing that the mega-domain structure remains for at least 5 days into the reprogramming process.

Unexpectedly, we could observe the distinct checkerboard pattern associated with genomic compartment structures[4] on the Xi in NPCs, which prompted us to further investigate the possible compartmentalization of the Xi. We applied PCA on our Hi-C data and initially found that the first eigenvector (PC1 values) captured the two mega-domains of the Xi (Fig. 2b top) as previously reported[7]. However, we reasoned that the dominant mega-domain boundary at *Dxz4* may obscure an underlying compartment structure, and therefore repeated this analysis on each mega-domain separately. Strikingly, this revealed an underlying compartment structure on the inactive $X^{mus}$ in NPCs (Fig. 2b), as originally suggested by Darrow et al.[10]. We identified ~75 A/B-like compartments on the Xi, which visually resembled the ~90 A/B compartments on the Xa in NPCs, which themselves were distinct from the ~120 A/B compartments of the two active X chromosomes in ESCs. Furthermore, clustering using UMAP showed that the compartment structure of the Xa had already been partially remodelled at D5, in accordance with the genome-wide restructuring of A/B-compartments during the transition from a somatic to a pluripotent state[34] (Fig. 2c). On the contrary, we found that both compartmentalization of the Xi, as well as strength of the mega-domain boundary, remained stable at D5 (Fig. 2c, d and Supplementary Fig. 2e). This suggests that the repressive chromatin state of the Xi might delay the remodelling of its compartment structure.

Next, we wanted to address if compartmentalization of the Xi was observable in our datasets due to the increased Hi-C resolution and G1-sorting or if it could have been observed in other datasets as well. We re-analyzed allele-specific in situ Hi-C data from NPCs[8] and mouse embryonic fibroblasts (MEFs)[9], and found that both exhibited compartmentalization of the inactive X (Supplementary Fig. 2f). We noted however that the data seemed visually noisier, suggesting that both high resolution and G1 sorting contributed to revealing the compartment structure of the Xi. In summary, this indicates that the A/B-like compartmentalization is not merely restricted to our differentiation system or cell type, but is a general property of the inactive mouse X chromosome.

While we were able to unveil an underlying compartment structure on the Xi, it remained possible that the A/B-like compartments identified within the dominant mega-domain structures were similarly attenuated as TADs[8] on the Xi and were hence diminished in their ability to spatially separate the chromosome. We, therefore, measured the overall interaction strengths within and between A-like and B-like compartments and visualized these differences in compartmentalization using saddle plots (Fig. 2e). We further computed the compartmentalization strength as a means to assess the degree of A/B spatial separation[34]. While we observed that compartmentalization strength of the active $X^{cas}$ in NPCs was higher than that of the inactive $X^{mus}$, we found overall that the compartmentalization

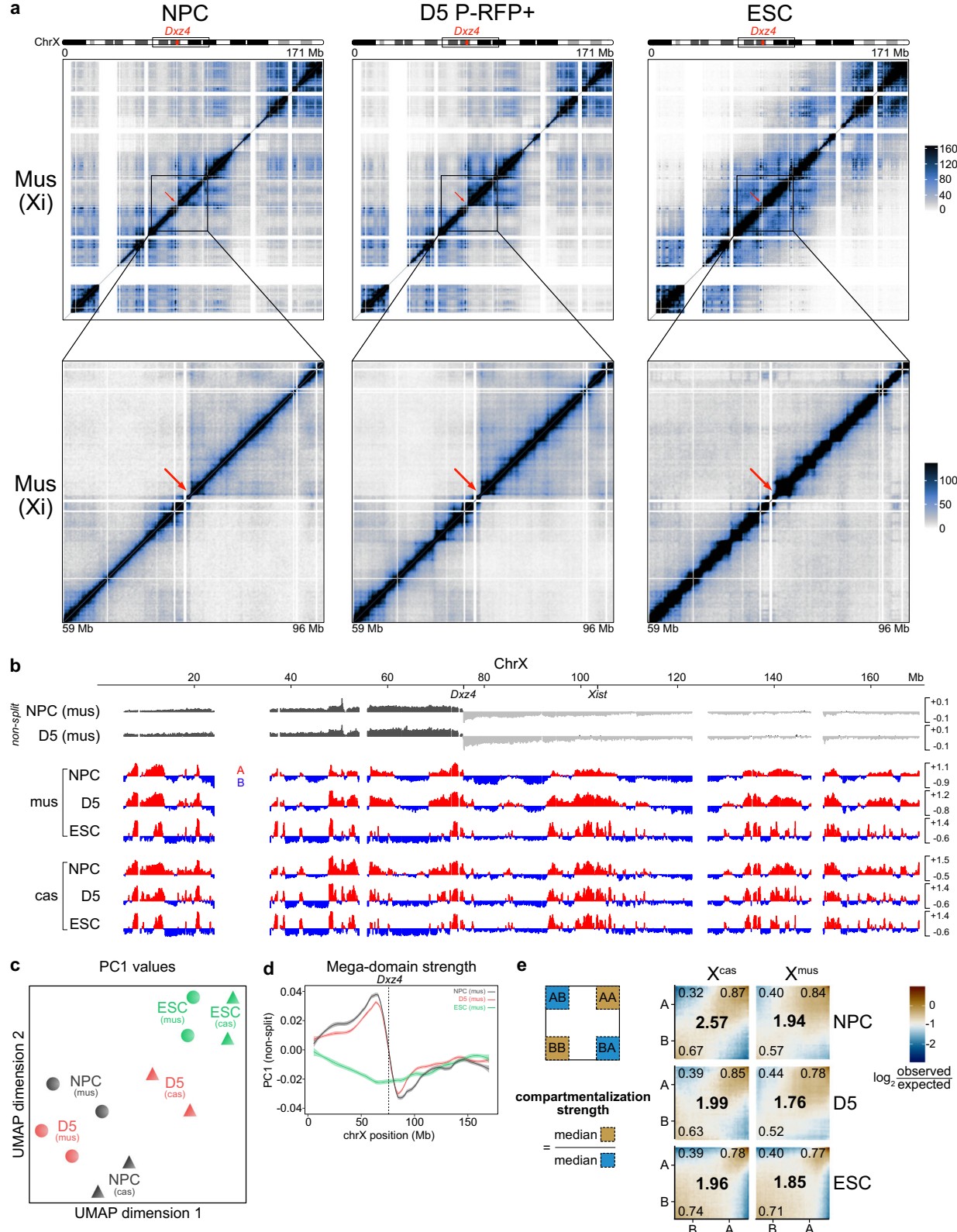

strength of the inactive X$^{mus}$ in NPCs is comparable or even higher than on D5 or in ESCs, therefore confirming our observation of the compartmentalization of the inactive X.

In summary, we have unveiled a previously overlooked A/B-like compartment structure on the inactive X chromosome, which is stably maintained during reprogramming until the X becomes reactivated.

**Subcompartmentalization of the inactive X.** In light of our unexpected discovery of an A/B-like compartment structure on the inactive X, we considered the possibility that further, more fine-grained levels of structural organization might exist. High-resolution Hi-C maps have enabled the discovery of an additional layer of organization that splits A/B compartments into 5 subcompartments[14]. These were not only shown to exhibit

**Fig. 2 The inactive X chromosome exhibits A/B-like compartmentalization. a** Allele-specific Hi-C maps of chromosome X$^{mus}$ at the inactive state in NPCs (left), intermediate state during reprogramming in D5 P-RFP+ cells (middle) and in the active state in ESCs (right). Top: Entire chromosome is shown at 200-kb resolution. Bottom: Zoom-in of the mega-domain boundary is shown at 100-kb resolution. Scale is shown in mega-bases (Mb). The mega-domain boundary *Dxz4* is indicated by a red arrow. White-shaded areas, unmappable regions. **b** A/B compartments of chromosome X at 100-kb resolution obtained with principal component analysis of matrices split at the *Dxz4* mega-domain boundary. Positive PC1 values represent A-like compartments (red); negative PC1 values represent B-like compartments (blue). Top: when matrices are not split at the *Dxz4* mega-domain boundary, then the PC1 corresponds to the two mega-domains for the inactive X chromosome. **c** Clustering of PC1 values to compare A/B-like compartmentalization of the X$^{mus}$ and X$^{cas}$ at different stages using UMAP (uniform manifold approximation and projection) ($n = 1,406$ bins). **d** Mega-domain strength on X$^{mus}$ depicted by the PC1 of the Hi-C matrices without splitting at *Dxz4*. Lines show smoothed mean from a fitted loess curve with span 0.25. Shading denotes 95% confidence interval. Dotted line indicates position of *Dxz4*. **e** Saddle plots showing the interactions within (AA, BB) and between (AB, BA) compartments (small numbers in the corners) of chromosome X. Data are presented as the log2 ratio of observed versus expected aggregated contacts between bins of discretized eigenvalues (50 categories, bin size = 100 kb). Overall compartmentalization strengths for X$^{mus}$ and X$^{cas}$ at different stages are shown as large numbers in the centre.

distinct interaction patterns but additionally displayed specific patterns of chromatin modifications[14], nuclear positioning[56,57], and chromatin interaction stability[58].

To investigate if such a subcompartmentalization exists on the inactive X chromosome as well, we utilized a previously reported approach[59] to segment our allelically-resolved Hi-C matrices into spatial clusters based on their intra-chromosomal interaction pattern (Fig. 3a). We used matrices of the inactive X$^{mus}$ of D5 P-RFP+ pre-iPSCs to define spatial clusters, as they capture the specific transitory stage structure of reprogramming (see below). Our spatial segmentation yielded 12 clusters, 5 on the left and 7 on the right mega-domain (Fig. 3b), with an average size of 317 kb for contiguous domains. Moreover, interaction patterns of these clusters were highly similar in NPCs, on both the inactive X$^{mus}$ and the active X$^{cas}$ (Supplementary Fig. 3a), in line with their overall resemblance in A/B-like compartment structure (Fig. 2b, c). We then consolidated these clusters into 5 sub-compartments based on their mutual interaction patterns (Fig. 3a) and their PC1 values in NPC X$^{mus}$ (Fig. 3c), revealing two A-like subcompartments (A1 and A2), one intermediate subcompartment we called AB-like and two B-like subcompartments (B1 and B2) (Fig. 3d). Similarly to subcompartments identified globally in human cells[14], we found subcompartments A1 and A2 to be the most gene-rich on the inactive mouse X chromosome (Fig. 3e). Accordingly, we also found A-like subcompartments to be enriched in SINE repeats (Supplementary Fig. 3b), whereas LINE1 elements were preferentially enriched in B-like subcompartments (Supplementary Fig. 3c)[60].

Because Xist initially targets gene-rich regions during XCI[61], we asked how such preferential Xist enrichment would be reflected across subcompartments on the inactive X in NPCs. To this end, we integrated published Xist CHART-seq and ChIP-seq data sets obtained from NPCs[8] with our compartment data. We observed a differential enrichment of Xist RNA along the subcompartments, with the highest levels of Xist, detected in subcompartment A1, which harbours the *Xist* locus itself, and then gradually decreasing in the other compartments towards reaching the lowest levels in B2 (Fig. 3b, f). Considering this differential Xist enrichment, we wondered if this would lead to a distinct epigenetic makeup of subcompartments. In line with previous reports showing polycomb-recruitment to the inactive X and Xist RNA spreading to be interdependent[18,62–66], we found H3K27me3 to be enriched in A-like subcompartments (Fig. 3g). On the contrary, H3K9me2-associated protein CBX1 was enriched in B-like subcompartments (Fig. 3h), which was expected considering that H3K27me3 and H3K9me2/3 were shown to occupy distinct subcompartments on autosomes[7,11,13,14]. Moreover, in light of previous work demonstrating repulsion of the architectural proteins CTCF and the cohesin RAD21 by Xist from the inactive X[11], we observed a significant reduction of CTCF in A-like subcompartments, while

no clear trend was observed for RAD21 in our analysis (Supplementary Fig. 3d, e).

An intrinsic property of chromosomal compartments is the spatial segregation of distinct chromatin states[4]. Having unveiled subcompartments on the inactive X with specific epigenetic signatures, we therefore wondered how this may shape the overall structure of the X chromosome. We reasoned that utilizing the intra-chromosomal interaction pattern of these clusters would allow us to deduce their spatial relation with each other. We used ForceAtlas2[67], to construct a force-directed network using either NPC or D5 P-RFP+ X$^{mus}$ matrices, where each cluster has been consolidated into a single node (Fig. 3i and Supplementary Fig. 3f). This analysis revealed a spatial organization with clusters of B-like subcompartments occupying the exterior of the chromosome, with a generally low degree of connectivity. Clusters of subcompartment AB were found to be situated at the interface between A- and B-like clusters, with A-like clusters occupying the centre of the network. Moreover, A1-like clusters 5 in the left and 6 in the right mega-domain appear to reside in a spatial location where the two mega-domains come closest to each other. To confirm this observation, we quantified the degree of 3D-interactions bridging the mega-domain boundary. Indeed, we found A1-like clusters (5 and 6) to exhibit the highest degree of inter mega-domain interactions (Fig. 3j and Supplementary Fig. 3g). It is of interest to note that the *Xist* locus resides within cluster 6 (Fig. 3b). Therefore the close 3D-proximity between clusters 5 and 6 may facilitate the efficient spreading of Xist RNA within the gene-rich A1 compartment, which occurs first during X-inactivation[61,68]. Xist RNA and the distinct epigenetic status of the A-like and B-like domains could subsequently contribute to the maintenance of the stable underlying compartment structure on the Xi as suggested previously[9], in contrast to the variable cell-type-specific A/B-compartments on the Xa (Fig. 2c).

In summary, we have unveiled a subcompartment structure on the inactive X that is characterized by preferential binding of Xist RNA to gene-rich A-like compartments. Moreover, although being separated by the mega-domain boundary, gene-rich compartments are in close spatial proximity, suggesting that the 3D-structure of the X chromosome could provide a scaffold for efficient Xist-mediated gene silencing and dosage compensation.

**3D-clustering of escapees and early X-reactivation.** Our identification of a fine-grained subcompartment structure on the inactive X prompted us to ask how this might impact gene expression and chromatin accessibility. Specifically, we wanted to know if the dynamics of X-reactivation might be influenced by these structural features.

When we assessed the allelic expression of the 12 spatial clusters, it became apparent that genes escaping X-inactivation in

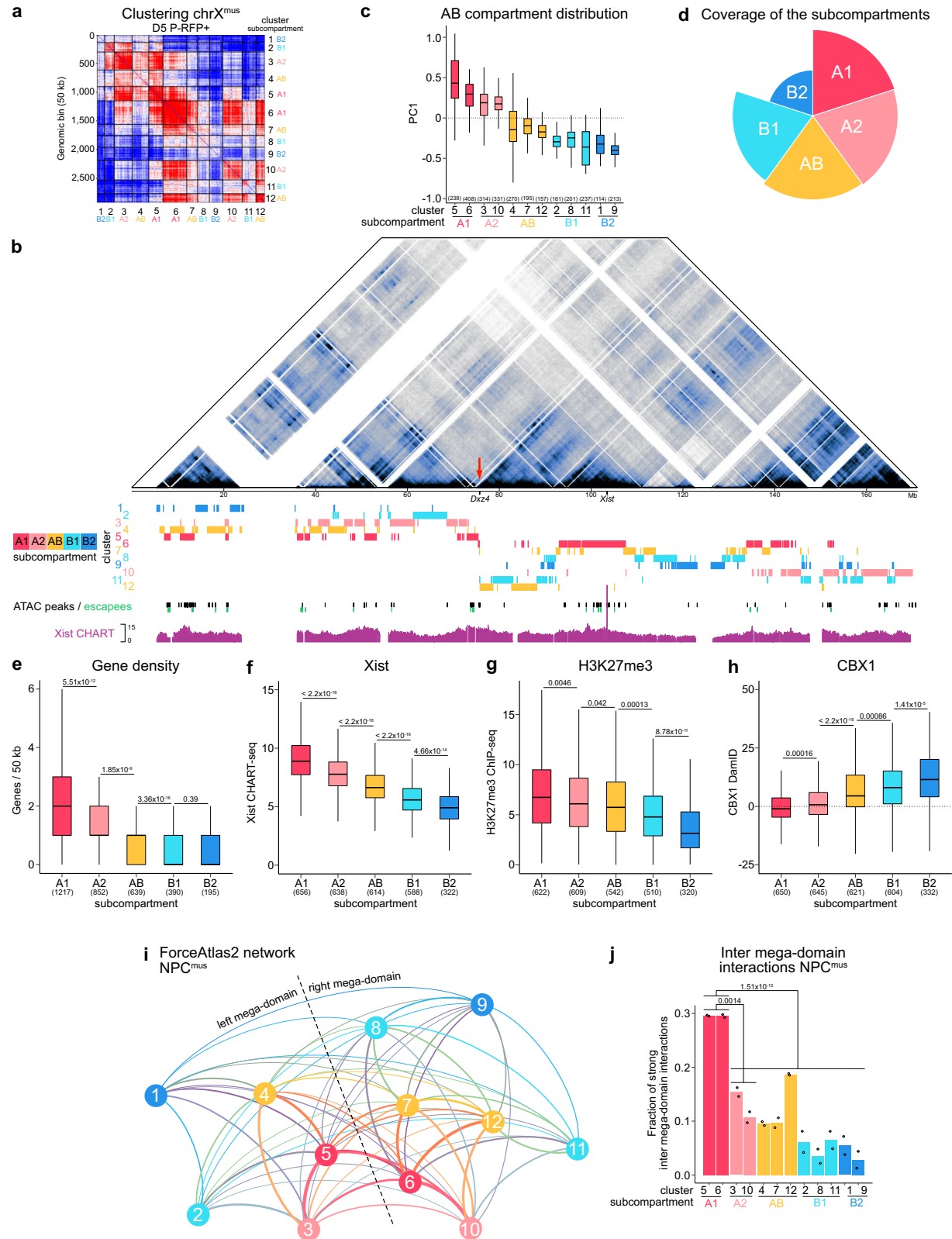

NPCs (Supplementary Fig. 4a) almost exclusively resided in subcompartment A1 (Fig. 4a), with up to 30% of genes in cluster 5 being escapees. Moreover, while on the active X^cas chromatin accessibility was generally higher in the A-type compartment compared to the rest, we found on the inactive X^mus chromatin accessibility specifically of cluster 5 to be at much higher levels when compared to the rest of the clusters (Fig. 4b).

These observations motivated us to ask if this would advance the timing of chromatin opening and gene reactivation in cluster 5. Indeed, when we assessed the dynamics of chromatin opening of the Xi (X^mus) during reprogramming, we found early chromatin opening at days 4 to 6 to specifically occur in cluster 5 (Fig. 4c, d). However, early chromatin opening of cluster 5 was restricted to around 25% of iPSC levels, with the most significant

**Fig. 3 Subcompartmentalization of the inactive X chromosome. a** Identification of spatial clusters (numbers 1 to 12 on the axes) and their associated subcompartments (text in colour next to cluster labels) on the inactive X using *k*-means clustering on a balanced matrix of chromosome X$^{mus}$ D5 P-RFP+ at 50-kb resolution. Red areas interact more while blue areas interact less. **b** Allele-specific Hi-C map of chromosome X$^{mus}$ in NPCs at 100-kb resolution. Scale is shown in mega-bases (Mb). Mega-domain boundary *Dxz4* is indicated by a red arrow. White-shaded areas, unmappable regions. Position of spatial clusters is shown below. Position of ATAC peaks in NPC X$^{mus}$ is shown in black, genes escaping X-inactivation in NPCs are shown in green. Xist RNA binding pattern in NPCs (CHART-seq, composite scaled tracks) taken from ref. [8]. **c** Distribution of PC1 values in NPC X$^{mus}$ of the spatial clusters. Box plots depict the first and third quartiles as the lower and upper bounds of the box, with a band inside the box showing the median value and whiskers representing 1.5x the interquartile range. n is given in brackets and indicates number of 50 kb bins. **d** Polar chart showing the coverage of the subcompartments on chromosome X (fraction of linear sequence occupied by each subcompartment). **e** Gene density of subcompartments as number of genes per 50 kb bin. Sample sizes are given in brackets. **f** Xist RNA enrichment of subcompartments in NPCs (composite scaled data). CHART-seq data from ref. [8]. **g** H3K27me3 enrichment of subcompartments in NPC$^{mus}$. ChIP-seq data from ref. [8]. **h** CBX1 enrichment of subcompartments in NPC$^{mus}$. DamID data from ref. [8]. **e–h** The numbers above the bars indicate *p*-values (two-sample unpaired Wilcoxon-Mann-Whitney test with R defaults). Box plots depict the first and third quartiles as the lower and upper bounds of the box, with a band inside the box showing the median value and whiskers representing 1.5x the interquartile range. n is given in brackets and indicates number of 50 kb bins. **i** Network of spatial clusters on chromosome X$^{mus}$ in NPCs obtained by applying the ForceAtlas2 algorithm to Hi-C interaction patterns of spatial clusters. Each cluster represents a single node of the network. Line width correlates with interaction strength. **j** Inter-mega-domain interactions of clusters (across the mega-domain boundary) in NPC$^{mus}$. The numbers above the bars indicate *p*-values (unpaired two-samples t-test with R defaults). *n* = 2 biologically independent replicates.

opening at time point D6 X-GFP+, like all the other clusters (Fig. 4c and Supplementary Fig. 4b). Similarly, when we assessed gene reactivation dynamics of the gene-rich A-like clusters (Fig. 4e and Supplementary Fig. 4c), we observed about one-quarter of cluster 5 genes to reactivate early. Analogous to chromatin opening, early gene reactivation was restricted to around 25% of iPSC levels (Supplementary Fig. 4d). However, while we intended to ensure that the isolated subpopulations are homogeneous, due to the analysis of bulk data, we cannot exclude the possibility that our observations are attributable to a full reactivation of these genes in only a subset of cells.

Why do early chromatin opening and early gene reactivation happen almost exclusively in cluster 5? First, we hypothesized that higher absolute expression levels in NPCs might aid earlier reactivation. However, when we compared the expression levels of early and main reactivating genes on X$^{mus}$ in NPCs (Supplementary Fig. 4e), we found no significant differences. The differences appeared only later and were clearly visible at D4 (Supplementary Fig. 4f). Next, we asked if early reactivating genes might be bound by a distinct set of transcription factors expressed early during reprogramming. We set out to identify enriched transcription factor binding motifs using the MEME suite[69], comparing differential ATAC-seq peaks of early reactivating genes at D4 P-RFP+, to main reactivating genes at D6-RFP+ (Supplementary Fig. 4g). However, we could not detect any significantly enriched differential motifs (Supplementary Fig. 4g), suggesting that binding of specific transcription factors is unlikely to be the main driver in directing early gene reactivation.

As we showed that cluster 5 harbours the highest percentage of escapee genes, we considered that close distance to escapees, and therefore also close vicinity to open regions, might facilitate early reactivation as shown previously[32]. Indeed, we found that overall, genes in cluster 5 were in closest proximity to escapees (Fig. 4f). Compellingly, when we specifically compared the distance to escapees for early and main reactivating genes within cluster 5, we found that early genes were significantly closer to escapees than the rest of genes in this cluster (Fig. 4g). Additionally, we observed a significant enrichment of SINE elements near promoters of escapees and early reactivated genes in cluster 5 (Supplementary Fig. 4h), a property previously described for escapees and genes prone for reactivation after Xist-depletion[70,71]. Moreover, while we did not detect increased expression levels of early genes in NPCs, where they were still transcriptionally inactive (Supplementary Fig. 4e), we did find promoters of early reactivating genes to already be more accessible in NPCs (Fig. 4h). This is in line with our observation,

that chromatin opening at gene promoters precedes transcription during the initiation of X-chromosome reactivation (Supplementary Fig. 4i–k).

In summary, partial reactivation of genes early on during reprogramming is confined to a distinct spatial cluster that is characterized by a high number of genes escaping X-inactivation.

**Remodelling of the X-inactivation centre leads to *Xist* down-regulation.** A critical event during X-reactivation is the down-regulation of *Xist*[25,33], which coincides with the upregulation of the naive pluripotency network (Fig. 1e, f) and with the main occurrence of chromatin opening and reactivation of X-linked genes (Figs. 1d and 4c). In order to derive mechanistic insights into the *Xist* downregulation process during reprogramming, we examined changes of *Xist* and its known regulators at the X-inactivation centre (*Xic*) (Fig. 5 and Supplementary Fig. 5).

We first focussed on the chromatin status of the *Xist* locus itself (Fig. 5a, b and Supplementary Fig. 5a). Specifically, we observed a reduction in accessibility at the main *Xist* promoter 1, which preceded the full downregulation of Xist RNA during reprogramming and the loss of the Xist RNA cloud (Supplementary Fig. 5b, c). Following the opposite trend, we saw a gain in accessibility at *Xist* intron 1, a known binding hub for pluripotency factors such as OCT4, SOX2, NANOG, and PRDM14[25,35,49–51]. Like Xist RNA downregulation, gain in accessibility at *Xist* intron 1 took place in two phases: The first step occurred around D4, with a strong accessibility gain at *Xist* intron 1 and a two-fold reduction in Xist RNA levels, presumably due to expression of the reprogramming cassette, as well as the reactivation of endogenous *Oct4* expression (Fig. 1f). This was followed by a gradual increase in accessibility from D4 on, following the expression of naive pluripotency genes such as *Nanog*, *Zfp42/Rex1* and *Prdm14*, in line with their known role in repressing *Xist* in ESCs or during iPSC reprogramming[25,50,51] (Figs. 1e and 5b; and Supplementary Fig. 5b).

To gain further insights, we next investigated changes in the regulatory landscape of the entire *Xic*. Structurally, the *Xic* is divided into two functionally opposing domains[6,72–75]: TAD-D, which contains the non-coding genes *Tsix*, *Xite* and *Linx*, which are repressors of *Xist*; and TAD-E, which harbours *Xist* itself and its activators, the non-coding genes *Jpx* and *Ftx* and the protein-coding *Rlim/Rnf12* (Fig. 5c).

We first assessed changes in 3D-organisation and noticed a gradual strengthening of the TAD border between TAD-D and TAD-E during reprogramming, as indicated by a drop in the insulation score (Fig. 5c). While we could not detect major

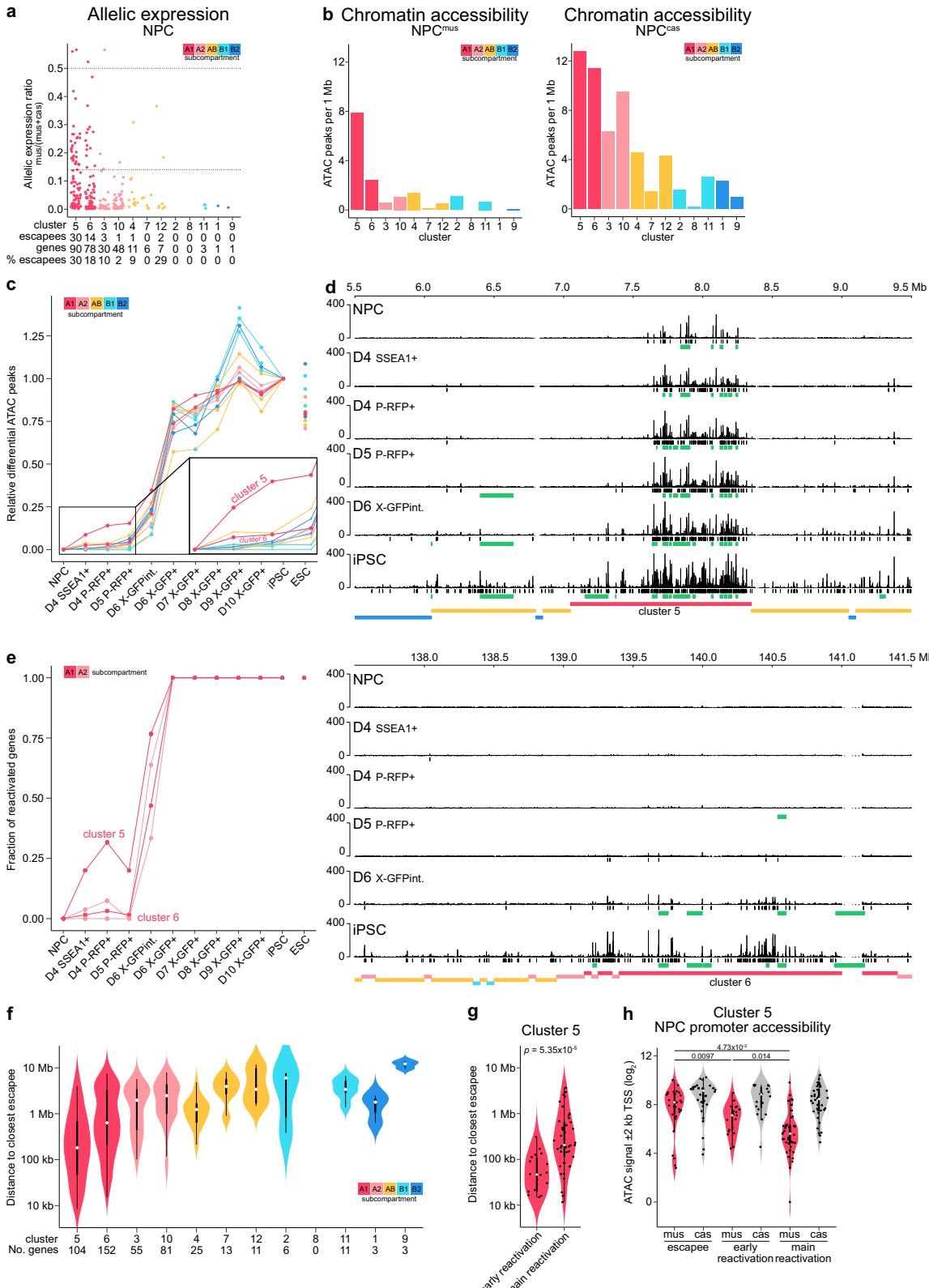

changes in the 3D-organisation of TAD-D on the inactive $X^{mus}$ at D5 (Fig. 5d–f) despite seeing them on the active $X^{cas}$ (Supplementary Fig. 5d–f), we, on the contrary, noted various changes of TAD-E early on. We observed a decrease in 3D-interactions on the Xi ($X^{mus}$), specifically within the region spanning *Xist*, *Jpx*, and *Ftx* already on D5 preceding X-reactivation, while we saw an increase in interactions between

*Ftx* and *Rlim* (Fig. 5d–f). This shows that while restructuring of TAD-D only occurs after X-reactivation, suggesting it to be a consequence rather than a cause of *Xist* downregulation, changes in TAD-E precede and therefore more likely play a role in X-reactivation.

Next, we took advantage of our high-resolution RNA-seq data set, to deduce putative roles of *Xic* genes in *Xist* regulation during

**Fig. 4 Initiation of chromatin opening and gene expression from a distinct 3D cluster. a** Allelic expression ratio (= mus/(mus+cas)) of X-linked genes in spatial clusters. Cutoff >0.14 defines escapees (Supplementary Fig. 4a). For biallelic expression, ratio = 0.5. Only protein-coding genes with sufficient allelic information and expression for chromosome X$^{cas}$ are counted (see methods). **b** Chromatin accessibility of each spatial cluster in NPCs shown as number of ATAC peaks per 1 Mb. **c** Dynamics of chromatin opening of spatial clusters. Only new peaks differential from NPCs were used. Relative differential ATAC peaks were then obtained by dividing the sum of peaks of each cluster at a given time point, by the sum of peaks in iPSC. Therefore NPCs will have a value of 0 and iPSCs a value of 1. Zoom-in shows early chromatin opening from NPCs until D6 P-RFP+. **d** ATAC-seq profiles of chromatin opening at two representative X-linked regions of 4 Mb. Position of ATAC peaks is shown in black (except for NPCs, differential new peaks compared to NPCs are shown). Genes either escaping X-inactivation in NPCs or being reactivated based on RNA expression are shown in green. Position of spatial clusters is shown at the bottom. **e** Dynamics of gene reactivation of gene-rich A-like clusters. Fractions of reactivated genes per cluster are shown. 0, no reactivated gene. 1, all genes reactivated. Threshold for gene reactivation, allelic expression ratio >0.14. **f** Violin plots showing the linear distance of genes to the closest escapee. Distances were calculated between the transcriptional start sites (TSS). Numbers of genes per cluster are given at the bottom. **g** Violin plots showing the linear distance to the closest escapee for genes of cluster 5 reactivating early, at D4 P-RFP+, compared to genes reactivating after that ("main reactivation"). Distances were calculated between the TSS. **h** Violin plots showing the promoter accessibility of genes of cluster 5. ATAC signal in a window of ±2 kb around the TSS was summed. **g**, **h** The numbers above the plot indicate *p*-values (two-sample unpaired Wilcoxon-Mann-Whitney test with R defaults).

X-reactivation. As a mirror image to the downregulation of *Xist*, we observed the upregulation of *Tsix* (Supplementary Fig. 5b), the antisense repressor of *Xist* during X-inactivation[39,41,76]. Since we have been using a functionally null *Tsix* truncation (TST) allele on the X$^{mus}$ in our study[40,41], we confirmed previous findings that *Tsix* is dispensable for *Xist* downregulation during X-reactivation in iPSCs[25,77]. Furthermore, the kinetics in expression changes of *Linx*, as well as of the *Xist* activators *Ftx*[78] and *Rlim*[50,79] (Supplementary Fig. 5b), did not show a clear correlation with *Xist* downregulation, therefore making them unlikely to be the main regulators of *Xist* in our system. However, *Jpx*, which interacts with *Xist in cis*[72] and facilitates *Xist* expression during X-inactivation[80,81], was downregulated with a highly similar profile to *Xist* during reprogramming (Fig. 5g and Supplementary Fig. 5b). This suggests that *Jpx* downregulation might be a key step for decreasing Xist RNA levels during X-reactivation, which is in line with the suggested role of *Jpx* in facilitating *Xist* expression during X-inactivation maintenance[82].

In light of these results, we asked if reducing levels of Jpx RNA would be sufficient to lead to a downregulation of Xist RNA in our system. We therefore performed LNA-GapmeR-mediated knockdown of Jpx and Xist RNA in NPCs and subsequently analysed their expression by quantitative RT-PCR (Fig. 5h). We observed that both LNAs could robustly deplete their respective target transcripts. Moreover, whereas depletion of Xist did not affect levels of Jpx, knockdown of Jpx led to a significant reduction in levels of Xist RNA, confirming that Jpx RNA is required to maintain full *Xist* expression in post-XCI cells[82], including our NPCs.

We conclude that the *Xic* is remodelled during reprogramming at multiple levels, leading to downregulation of *Xist*, a critical step for X-reactivation. At the level of the *Xist* locus itself, we found changes in chromatin accessibility at the pluripotency factor-bound *Xist* intron 1 and the *Xist* promoter to precede X-reactivation. Furthermore, we observed early structural changes in *Xist* TAD-E, where we detected an early loss of regulatory contacts in between *Xist* and its activators *Jpx* and *Ftx*. Finally, the concurrent downregulation of *Jpx* and *Xist*, as well as Jpx RNA's role in maintaining *Xist* expression in NPCs, highlights *Jpx* downregulation as a candidate mechanism for *Xist* downregulation during X-reactivation.

**TAD formation in the absence of chromatin opening and reactivation of transcription.** The relationship and interplay between chromosome architecture and transcription during development has been an area of intense debate[59,83]. Whereas some evidence suggests that transcription determines 3D chromatin organization[37], it has been shown previously, that formation of TADs during zygotic genome activation is independent of

transcription, and not merely a consequence of it[38,84,85]. While there might be context-dependent differences, the inactive X illustrates a unique instance, where TADs are actively attenuated[8] by the action of the non-coding Xist RNA and are fully regained during X-reactivation. To get more mechanistic insights, we thus wanted to ask if the formation of structural domains on the inactive X precedes or rather follows chromatin opening and gene reactivation.

When inspecting Hi-C matrices on D5, we noticed a unique intermediate structure of the inactive X (Figs. 2a and 6a). It showed typical Xi features such as the mega-domains and their associated super-loops[10] between *Dxz4* and *Firre*, as well as between *x75* and *Dxz4* (Supplementary Fig. 6a), but also already emerging TAD structures typical of an active X (Fig. 6a). To assess the gain of TADs during X-reactivation on a quantitative level, we computed the insulation score[86] to assess the strength of TAD borders, and the domain score[53] to quantify the degree of connectivity within TADs. In contrast to compartments, we already noticed changes in both insulation score and domain score in D5 P-RFP+ cells (Fig. 6b–d). Specifically, we observed a strengthening of TAD borders (Fig. 6b) and an increase in the range of the insulation score (Fig. 6c and Supplementary Fig. 6b). Moreover, we detected a significant increase in the domain score of the Xi (X$^{mus}$) (Fig. 6d and Supplementary Fig. 6c), revealing an increased connectivity of TADs. Our observation of preferential binding of Xist RNA to gene-rich A-like subcompartments (Fig. 3f) and its role in repelling architectural proteins like CTCF and cohesin from the inactive X[11], prompted us to ask if this would lead to differential domain score dynamics among subcompartments. Indeed, when we assessed the relative domain score, to highlight changes occurring at D5, we observed that B-like subcompartments underwent the largest increase in TAD connectivity, while A-like compartments lagged behind (Fig. 6e and Supplementary Fig. 6d). Moreover, when we assessed domain score differences and correlated these to local enrichment of Xist RNA in NPCs, we found a strong anti-correlation between the levels of Xist, and the relative domain score at D5 (Fig. 6f). This suggests that high levels of Xist inhibit the early formation of TADs, in agreement with the delayed gain in TADs observed in generally Xist-rich A-like compartments when compared with Xist-poor B-like compartments. Furthermore, considering that levels of H3K27me3 highly correlate with levels of Xist RNA, we expectedly observed a similar anti-correlation between H3K27me3 levels and domain score dynamics (Fig. 6g). Importantly, we did not find that differential domain score dynamics of subcompartments were influenced by differences already present in NPCs, as we observed that the initial domain score in NPCs had little influence on the relative score at D5 (Supplementary

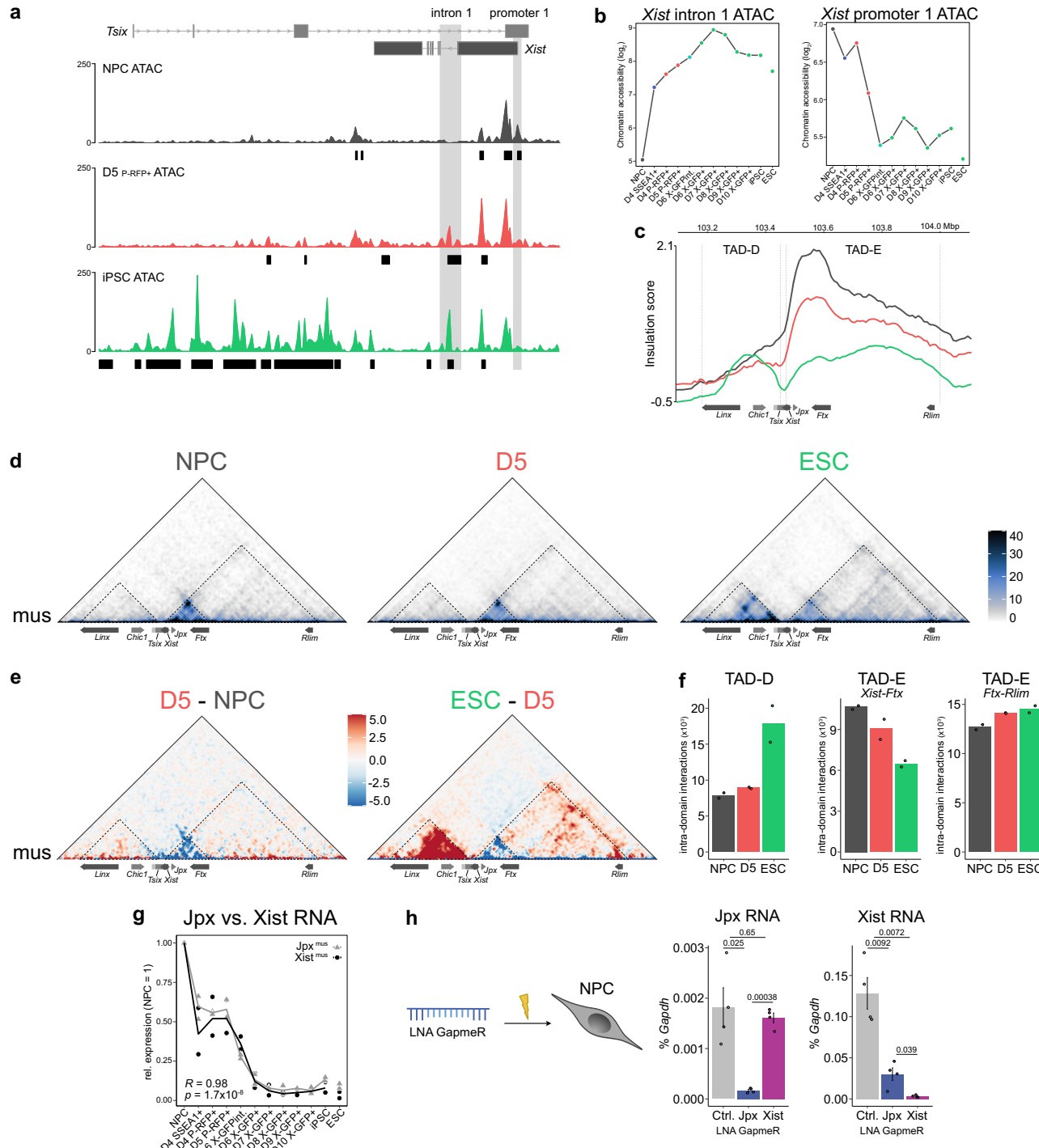

**Fig. 5 Remodelling of the X-inactivation centre leading to *Xist* downregulation. a** ATAC-seq profiles of chromatin opening on X^mus at a region encompassing the *Tsix* and *Xist* genes (mm10; 103,416,500 bp–103,490,000 bp). Position of ATAC peaks is shown in black (except for NPCs, differential peaks compared to NPCs are shown). **b** Chromatin accessibility on X^mus at *Xist* promoter 1 (mm10; 103,482,600 bp–103,483,800 bp) and *Xist* intron 1 (mm10; 103,470,900 bp–103,474,200 bp) as depicted in **a**. **c** Insulation score at 10-kb resolution at a region encompassing *Tsix* TAD-D (mm10; 103.18 Mb–103.45 Mb) and *Xist* TAD-E (mm10; 103.47 Mb–104.0 Mb). Dotted lines show TAD borders. Only genes with implicated roles in X-inactivation or X-reactivation are shown. **d** Allele-specific Hi-C maps of chromosome X^mus at 10-kb resolution at a region encompassing TAD-D (left) and TAD-E (right) in NPCs, D5 cells and ESCs. **e** Differential allele-specific Hi-C maps between D5 and NPCs (left), and ESCs and D5 cells (right) of the region shown in **d**. **d, e** Dotted lines show TAD borders and additionally separate TAD-E in two regions at the TSS of *Ftx* (mm10; 103.62 Mb) for quantification in **f**. **f** Sum of intra-domain interactions of X^mus are shown. TAD-E was separated in two regions at the TSS of *Ftx*. *n* = 2 biologically independent replicates. **g** Expression of Jpx^mus and Xist^mus relative to the levels in NPCs. *R* and *p*-values calculated by Pearson's correlation are shown. *n* = 2 biologically independent replicates. **h** NPCs at day 9 of differentiation were nucleofected with LNA GapmeRs and expression of Jpx and Xist analysed by quantitative RT-PCR one day later normalized to Gapdh. The numbers above the bars indicate *p*-values (unpaired two-samples *t*-test with R defaults). Error bars denote SEM. Ctrl., control LNA. *n* = 4 biologically independent replicates. Source data are provided as a Source Data file.

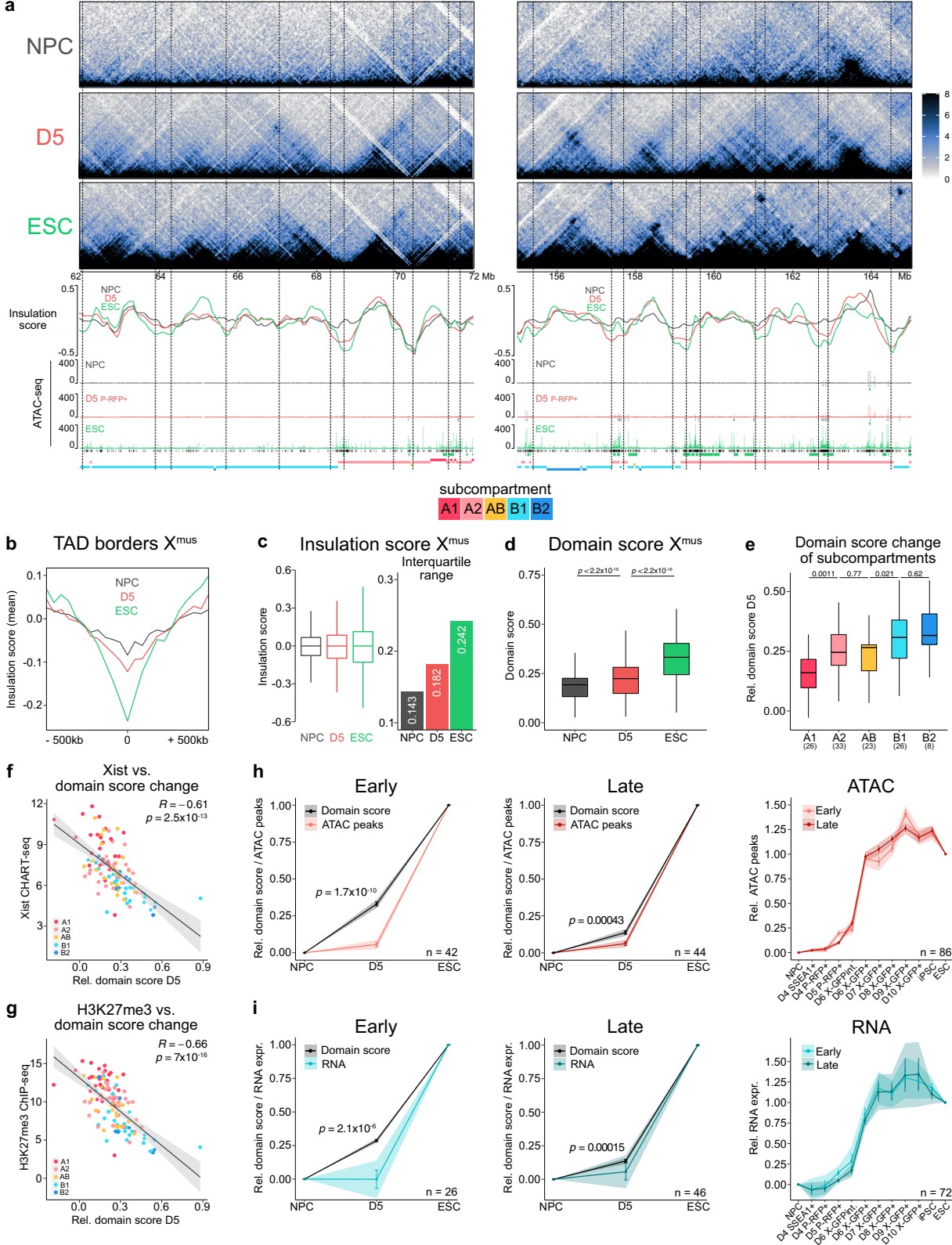

Fig. 6e). This confirmed that the domain score is a reasonable proxy for the timing of TAD appearance. Furthermore, we wanted to rule out that these observations arose from a heterogeneous cell population at D5, containing NPC-like cells (with mega-domains and no TADs), as well as ESC-like cells (with TADs and no mega-domains). Our observation that the strength of the mega-domain boundary remained stable at D5

(Fig. 2d), suggested that the vast majority of cells still harboured a mega-domain, assuming similar mega-domain strength in cells as done previously[19]. To address a possible heterogeneity of TADs, we in silico mixed the Hi-C maps of NPCs and ESCs at different ratios and measured the relative domain scores of subcompartments (Supplementary Fig. 6f). We found that the real domain scores at D5 were not compatible with a single mixing ratio.

**Fig. 6 Structural changes during X-reactivation in the absence of chromatin opening and transcription. a** Two representative X-linked regions of 10 Mb for early TAD formation are shown. Allele-specific Hi-C map of chromosome X$^{mus}$ at 20-kb resolution. Scale is shown in mega-bases (Mb). Insulation scores at 50-kb resolution, dashed line indicates cut-off for TAD borders at -0.086. ATAC-seq profiles with ATAC peaks shown in black (except for NPCs, differential peaks compared to NPCs are shown). Genes either escaping X-inactivation in NPCs or being reactivated based on RNA expression are shown in green. Position of subcompartments is shown at the bottom. **b** Meta region plot of insulation score at TAD boundaries at each time-point. Lines show mean. $n = 116$ TAD boundaries. **c** Comparison of insulation scores for chromosome X$^{mus}$. Interquartile range of insulation scores is shown on the right. $n = 2,785$ 50 kb bins. **d** Comparison of domain scores for chromosome X$^{mus}$. $n = 100$ TADs. **e** Degree of change in domain score of X$^{mus}$ of subcompartments on D5. The relative domain score at D5 = (D5-NPC)/(ESC-NPC). $n = 100$ TADs. **c–e** Box plots depict the first and third quartiles as the lower and upper bounds of the box, with a band inside the box showing the median value and whiskers representing 1.5x the interquartile range. **d, e** The numbers above the lines indicate $p$-values (two-sample unpaired Wilcoxon–Mann–Whitney test with R defaults). **f** Correlation between Xist RNA CHART-seq enrichment in NPCs[8] and the relative domain score at D5 is shown. Points represent TADs. Colours of points indicate subcompartments. $R$ and $p$-values calculated by Pearson's correlation are shown. Black line represents linear regression fitting. Shading denotes 95% confidence interval of the fit. **g** As **f** for H3K27me3 ChIP-seq in NPCs[8]. **h** Comparison of domain dynamics and chromatin opening. Relative domain score is shown. Relative sum of ATAC peaks per TAD is shown. Only TADs with a minimum of 15 peaks in ESC were used. Early, TADs that changed from NPC to D5 (Supplementary Fig. 6g). Late, TADs that did not change from NPC to D5. Line shows mean. Error bars denote SEM. Shading denotes 95% confidence interval. Right panel shows comparison of chromatin opening dynamics (relative ATAC peaks) between early and late TADs. **i** Comparison of domain dynamics and gene reactivation. Relative domain score is shown. Relative mean expression per TAD is shown. Early, TADs that changed from NPC to D5. Late, TADs that did not change from NPC to D5. Line shows mean. Error bars denote SEM. Shading denotes 95% confidence interval. Right panel shows comparison of gene reactivation dynamics (relative RNA expression) between early and late TADs. (**h, i**) The numbers above the lines indicate p-values (two-sample unpaired Wilcoxon–Mann–Whitney test with R defaults). $n$ indicates number of TADs.

---

While single-cell structural data would be necessary to directly resolve the temporal relationship of TAD formation and loss of the mega-domain structure, our data strongly suggest that the transitory structure observed at D5 is unlikely the result of a heterogeneous cell population and that therefore, TAD formation occurs in the presence of the two mega-domains.

Next, to identify TADs that have undergone domain score changes between stages, we performed $k$-means clustering on the relative domain score (Supplementary Fig. 6g), which showed that 55% of TADs already increased connectivity at D5 by more than 20% ("early", increase in domain score; "late", no increase in domain score). In line with our previous observations, we found B-like subcompartments to be enriched in early TADs, whereas TADs of subcompartment A1 were mostly designated late TADs (Supplementary Fig. 6h). When we then assessed chromatin opening at early TADs, we observed a significantly higher degree in domain score change, compared to an only mild increase in the number of ATAC peaks suggesting that TAD formation occurs before the appearance of chromatin opening (Fig. 6a, h). In agreement with this and our observation that chromatin opening precedes gene reactivation (Supplementary Fig. 4i–k), we also found that TAD formation occurred in the absence of significant changes in gene expression at D5 (Fig. 6i). Moreover, when we compared the kinetics of chromatin opening (Fig. 6h) and gene reactivation (Fig. 6i) between early and late TADs, we did not find any significant differences. Therefore we conclude that TAD formation does not necessarily direct chromatin opening and gene reactivation.

Taken together, we show that early changes in TAD connectivity initiate from B-like subcompartments on the inactive X and anti-correlate with the local presence of Xist RNA. Moreover, our data show that TAD formation during X-reactivation often precedes and occurs without significant chromatin opening and gene reactivation, while intriguingly early TADs do not open chromatin or reactivate genes before late TADs. This suggests that chromatin opening and transcription are not essential drivers of the structural remodelling of the X chromosome during X-reactivation or vice versa, illustrating the mechanistic independence between these two events.

## Discussion

The interplay of chromatin conformation and transcriptional activity during cell fate transitions has been a topic of intense

debate[83,87]. Here we addressed this question by introducing a tailor-made system to study X-chromosome reactivation during iPSC reprogramming. The high X-reactivation efficiency of the PaX system allowed us to perform Hi-C on a synchronous population of cells and to follow how the inactive X switches from an inactive heterochromatic state to an active euchromatic one. We thereby uncovered an underappreciated A/B-like compartment structure on the inactive mouse X chromosome, which resembles its active counterpart and separates distinct chromatin domains. We detect the first signs of X-reactivation to initiate from regions escaping XCI, while full reactivation of most genes occurred later, yet in a rapid fashion, coinciding with the downregulation of Xist RNA. TAD structures emerged during X-reactivation before apparent gene reactivation, suggesting that transcriptional and structural remodelling of the X chromosome are independently suppressed by Xist RNA, and therefore qualify as functionally distinct events.

Previous studies on X-reactivation dynamics during iPSC reprogramming were based on mouse embryonic fibroblast (MEF) reprogramming systems[25,32,33]. These suffered from low X-reactivation efficiencies and high sample heterogeneity, with only a small fraction of cells at a given time point being poised to undergo X-reactivation. This resulted in slow and gradual X-reactivation kinetics, lasting over the course of several days, making it difficult to study its steps on a regulatory level.

We therefore developed PaX, a tailor-made iPSC reprogramming system based on a dual pluripotency and X-reporter mouse ESC line that allowed us to obtain a high-resolution time course of gene reactivation and chromatin opening during X-reactivation and intersected it with changes in 3D-chromatin structure. Our system allowed us to isolate large amounts of homogeneous cell populations poised for X-reactivation, which would subsequently progress with high efficiency through X-reactivation, enabling us to faithfully analyze the stepwise progression of X-reactivation during reprogramming (Fig. 1). Therefore, when analyzing allele-specific gene expression dynamics of the inactive X, we could demonstrate that X-reactivation in our system occurred rapidly in the time span of approximately 24 h, mirroring faithfully the kinetics of the X-reactivation process in vivo in mouse blastocysts[26].

A prominent feature of eukaryotic chromosomes is their spatial segregation into two compartments: A, corresponding to open chromatin and high mRNA expression, and B, corresponding to

closed chromatin and low expression[4], which manifest themselves as a distinct checkerboard pattern on Hi-C matrices. However, the mouse inactive X-chromosome has served as a unique exception to this observation, as it has been described to be devoid of A/B compartments and to be organized instead into two large mega-domains[7–9]. Here, we have uncovered that the mouse inactive X-chromosome is in fact segregated into A/B-like compartments that resemble the A/B structure on the active X within NPCs (Figs. 2 and 7a). While the high resolution of our Hi-C samples has facilitated this discovery, we show that this has been previously overlooked due to the strong features of the two mega-domains. The mega-domains are predominant when applying principal component analysis on Hi-C correlation matrices of the Xi, thereby obscuring the underlying compartment structure, which could only be unveiled by our separate analysis of the two mega-domains. Similarly, PCA on human chromosomes 4 and 5 initially was only able to capture the p and q arms, and only after splitting at the centromere, was the underlying A/B compartment structure revealed[4]. Moreover, our findings are supported by observations made in human cells, where compartments on the inactive X chromosome have been observed previously[10], and mouse primary neurons, where compartments have been suggested to exist on the Xi as well. Furthermore, when we applied our analysis strategy of

performing PCA separately for each mega-domain to published NPC and MEF in situ Hi-C data[8,9], we also observed A/B-like compartment structures on the Xi (Supplementary Fig. 2). This suggests that the underlying A/B-like compartmentalization is a general biological feature of the inactive X chromosome[10]. Importantly, we note that our observation of A/B-like compartments on a transcriptionally inactive chromosome favours a model where compartmentalization is not always driven by gene transcription.

The inactive X could therefore serve as a unique model to identify the underlying principles shaping compartment structures[88]. One notable aspect of the A/B-like compartments on the Xi is their distinct chromatin status, with A-like compartments being enriched in Xist RNA and the Polycomb-based H3K27me3 mark when compared to B-like compartments showing higher levels of CBX1, a reader of H3K9 dimethylation (Fig. 3). Nevertheless, except for escapees in the A-like compartment, both A-like and B-like compartments are transcriptionally inactive on the Xi. The Xi's A-like compartments differ from the classical A compartments present on the active X, which are active in transcription and enriched in H3K4me3, despite showing structurally similar interaction patterns. This could be explained by the fact that Xist RNA and Polycomb proteins during X-inactivation first enter into strongly

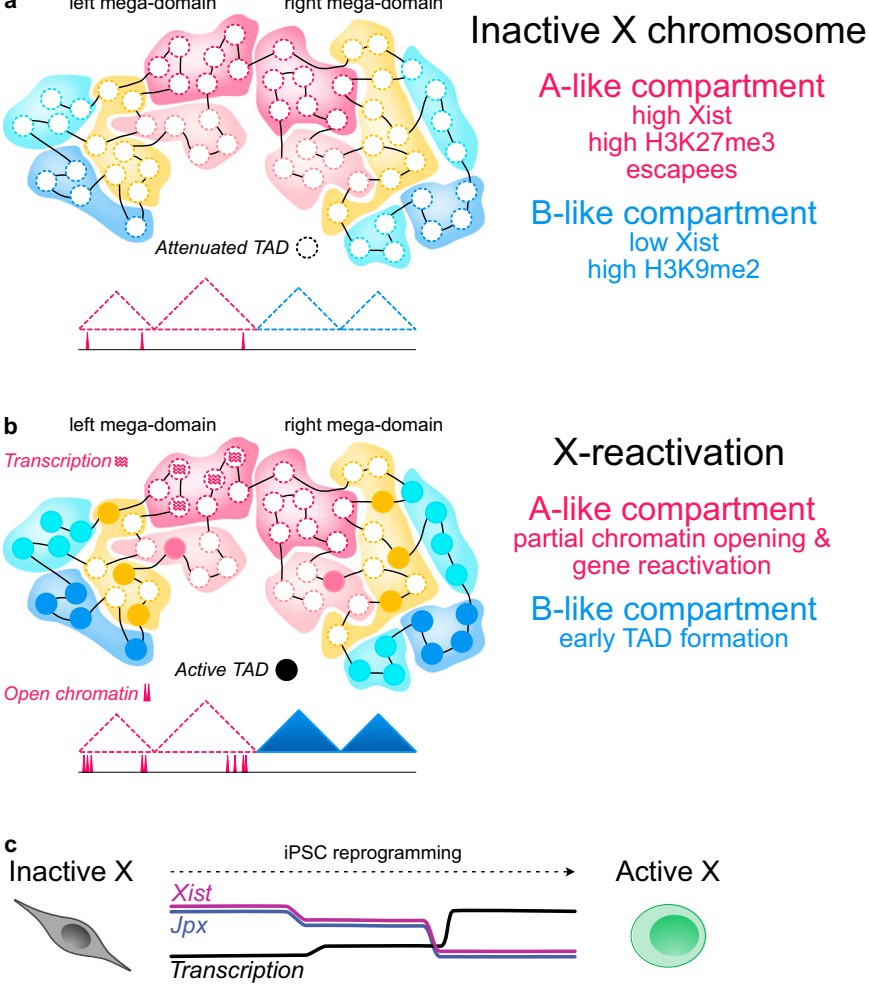

**Fig. 7 Graphical summary. a** The inactive mouse X chromosome displays an underlying compartment structure characterized by distinct epigenetic signatures. **b** Partial X-reactivation occurs first in an A-like subcompartment near escapee genes. Early formation of TADs occurs in B-like compartments, while mega-domains are still maintained, and precedes transcriptional reactivation. **c** Full X-reactivation then occurs rapidly in sync with *Xist* downregulation, which is promoted by downregulation of *Jpx*.

transcribed, gene-rich A-type regions[61,89], while only later and less efficiently also spreading into gene-poor, already silent B-type heterochromatin[90]. This makes sense from a functional point of view, where Xist-based silencing would be predominantly needed in the actively transcribed A-compartment to achieve X-linked gene dosage compensation while being less critical for gene-poor H3K9me2/3-marked heterochromatin in the B-compartment. As a consequence, Xist and its interacting partners might stabilize this structure, thereby establishing an epigenetic memory of the original A-compartment structure present at the time of X-inactivation, which is then maintained on the inactive X chromosome. The spatial separation between two types of heterochromatin on the inactive X, the Xist/Polycomb-rich A-like and the H3K9me-rich B-like heterochromatin, could be driven by liquid-liquid phase separation (LLPS) mechanisms, which have been described for both Polycomb[91] and H3K9me2/3 domains[92,93], thereby favouring a model where compartmentalization is driven by histone modifications as suggested previously[94]. Moreover, Xist RNA itself recruits a multitude of factors involved in LLPS to the inactive X[95] and Xist-deletion or depletion of Xist-associated LLPS factors like PRC1 and hnRNPK has been shown to significantly compromise the underlying Xi compartment structure[9]. Although A/B-like compartments of the inactive X, similar to the previously described S1 (A-like) and S2 (B-like) compartments[8,9], were thought to only exist transiently during X-inactivation, before subsequently being merged by SMCHD1, we were able to show that the compartmentalization persists on the inactive X and can be unveiled when applying separate analysis to the two-mega-domains. In support of our data, this dual heterochromatin structure has been previously observed on the human Xi[96], where it coincides with compartment structures as well[10], suggesting it to be a conserved property of the inactive X chromosome in mammals.

Underlying an A/B-like compartment structure, we have additionally discovered distinct spatial clusters and subcompartments on the sub-megabase level (Figs. 3 and 7a). Intriguingly, we found that gene-rich clusters, among others, are characterized by close spatial proximity and preferential binding of Xist RNA, arguing that spatial clustering of the inactive X provides a 3D-scaffold for efficient Xist-mediated gene silencing and dosage compensation. This is in line with observations that these gene-rich domains are the first areas to be coated by Xist RNA during the X-inactivation process[61].

Furthermore, we noticed that while the timing of complete X-linked gene reactivation was conserved in all subcompartments, partial X-reactivation early in reprogramming could be observed in a distinct spatial cluster, characterized by a high density in genes escaping X-inactivation (Figs. 4 and 7a, b), complementary to a previous observation that regions of high escapee density coincide with TADs in ESCs[12]. This suggested that proximity to accessible chromatin at escapees might facilitate further opening of regions nearby in a zipper-like fashion, leading to a partial basal reactivation of these genes. Indeed, we found this specific subset of early partially reactivated genes to lie in close proximity to escapees. However, the timing of complete X-reactivation of these genes was conserved compared to the rest of the genes. This suggests that while close distance to escapees, as reported previously[32], plays a role in X-reactivation, different mechanisms regulate the timing of complete X-reactivation, as discussed below.

An important event both for X-linked gene reactivation as well as for the structural remodelling of the X chromosome is the downregulation of Xist expression. Xist RNA has multiple distinct roles in establishing the silent chromatin state during X-inactivation based on its interaction with critical architectural and silencing factors like Cohesin, or SHARP/SPEN[11,15,16]. This

is in line with our observation that high Xist RNA occupancy on the inactive X anti-correlated with early TAD formation (Fig. 6 and Supplementary Fig. 6) and that gene reactivation kinetics were tightly linked to a sharp drop in Xist levels (Figs. 1, 5 and 7c). It is well appreciated that Xist is both, directly and indirectly, repressed by pluripotency factors such as OCT4, SOX2, NANOG, PRDM14, and ZFP42/REX1[35]. Indeed, we found that Xist intron 1, a known pluripotency factor binding hub[25,51], rapidly gained in accessibility with expression of the MKOS reprogramming cassette, coinciding with a partial decrease in Xist promoter accessibility and in Xist RNA levels early on during reprogramming (Figs. 1 and 5). This might allow the initial partial reactivation of genes lying near escapees, which we observed in structural cluster 5 (Fig. 4 and Supplementary Fig. 4). However, with the endogenous expression of naive pluripotency factors on D6 of reprogramming (Fig. 1), we saw a drastic drop in Xist expression, in line with NANOG, ZFP42 and PRDM14 being important Xist repressors[25,50,51]. As this coincided temporally with the full reactivation of X-linked genes, our data suggest that Xist downregulation, in addition to DNA demethylation and histone acetylation[32,33], is indeed a rate-limiting key step during X-reactivation thereby coupling it functionally to the reprogramming process.

Apart from trans-regulation by pluripotency factors, the Xist locus is regulated locally at the X-inactivation centre by the activators Jpx, Ftx, and Rlim within Xist TAD-E and repressors like Tsix, Xite and Linx within its neighbouring TAD-D[73]. When we assessed the topology of both TADs during reprogramming (Fig. 5 and Supplementary Fig. 5), we observed early structural changes to occur in particular within Xist TAD-E, suggesting it to be a driver of Xist downregulation. Intriguingly, we observed that downregulation of Jpx, a critical activator of Xist during the initiation[80,81] and maintenance[82] of X-inactivation, occurred with highly similar kinetics to Xist downregulation (Fig. 7c). Moreover, we found that knockdown of Jpx in NPCs was sufficient for the downregulation of Xist. This suggests that Jpx might therefore also play a role during X-reactivation. How Jpx itself is regulated remains an open question warranting further analysis.

Here we have shown that during X-reactivation, chromatin opening and gene reactivation are tightly linked (Fig. 4 and Supplementary Fig. 4). However, it was unknown whether changes in chromatin conformation are closely connected, as previously shown for autosomes during iPSC reprogramming[34]. We could demonstrate that, as the human Xi, the mouse inactive X chromosome features an A/B-like compartment structure (Fig. 2), and moreover found, consistent with previous findings[8], TADs to be strongly attenuated on the Xi (Fig. 6). This highlights that compartmentalization and TAD organization depend on distinct mechanisms[97,98] and that TADs need to be fully reestablished during X-reactivation. Intriguingly, a recent study on the dynamics of TADs during imprinted X-inactivation suggested that loss of TAD structure rather follows gene silencing and further showed that maintenance of TADs was restricted to escapee regions[36], which would propose a model where dynamics of TADs and transcription are intertwined on the Xi.

When we analyzed the Xi domain structure at day 5 of reprogramming, one day before the onset of full transcriptional reactivation, we found that more than 50% of TADs already displayed a significant increase in TAD connectivity (Fig. 6 and Supplementary Fig. 6). Considering our observation that mega-domains are still present at D5 (Fig. 2), this shows that TADs and mega-domains are distinct structural entities that are controlled independently. This is in line with the disappearance of TADs in the absence of mega-domain formation during imprinted X-inactivation[36] and the observation that mega-domains are dispensable for gene silencing and TAD attenuation during

X-inactivation[7,19,99]. Moreover, when we then compared these changes to the dynamics of chromatin accessibility and gene reactivation, we found that the connectivity of early TADs had already undergone significantly higher changes, compared to relatively mild changes in both chromatin accessibility and gene reactivation. This suggests that chromatin opening and transcription are not essential drivers of the structural remodelling of the Xi. Moreover, early TAD formation did not seem to prime for early reactivation either, as early and late TADs shared highly similar gene reactivation and chromatin opening dynamics, illustrating that these are mechanistically separate events, and that TADs are not required for transcription as shown previously[98]. However, a common denominator of both processes seems to be Xist. When we assessed the enrichment of Xist RNA, we noticed a strong correlation of early TAD formation with low Xist RNA occupancy, therefore early TADs forming mostly in Xist-poor B-like compartments and late TADs in Xist-rich A-like compartments (Fig. 7b). Considering the role of Xist in repelling architectural proteins like CTCF and cohesin from the inactive X[11,16], we propose that low levels of Xist facilitate the early restructuring of TADs, by allowing increased binding of CTCF and cohesin. Xist's gene silencing function however depends on a different set of binding partners than its structural role[11,15,16]. The removal of Xist-dependent repressive chromatin marks like H3K27me3 by UTX/KDM6A[26] and the gain in histone acetylation on the Xi after Xist downregulation[32], in combination with Xist-independent DNA-demethylation[33], have been shown to facilitate the transcriptional reactivation of the X. Therefore it is not surprising that this multistep process does not follow the same kinetics as the gain of TADs, although both processes are controlled by Xist, again highlighting its key role during X-reactivation.

Overall, our study provides mechanistic insight into the process of X-reactivation and the long-debated relationship of genome topology and transcription. We provide evidence that the mouse inactive X chromosome is in fact not as "unstructured" as it was believed to be and that, together with Xist downregulation, the fine structure of the inactive X parallels the reactivation kinetics. Moreover, our comprehensive dataset of the dynamics of transcriptional reactivation and chromatin opening and our high-resolution chromosome conformation maps of the reactivating X will provide a useful resource for future studies. Finally, our tailor-made PaX reprogramming system constitutes an optimized framework for further analysis of the chromatin dynamics and functional dissection of the X-reactivation process and, more generally, of the interplay between chromosome organization, chromatin architecture and gene regulation.

## Methods

**Embryonic stem cell culture**. Mouse embryonic stem cells (ESCs) were cultured on 0.2% gelatin-coated dishes in DMEM (Thermo Fisher Scientific, 31966021), supplemented with 10% FBS (ES-qualified, Thermo Fisher Scientific, 16141079), 1,000 U/ml LIF (ORF Genetics, 01-A1140-0100), 1 mM Sodium Pyruvate (Thermo Fisher Scientific, 11360070), 1x MEM Non-Essential Amino Acids Solution (Thermo Fisher Scientific, 11140050), 50U/ml penicillin/streptomycin (Ibian Tech, P06-07100) and 0.1 mM 2-mercaptoethanol (Thermo Fisher Scientific, 31350010). Cells were incubated at 37 °C with 5% CO$_2$. The medium was changed every day and cells were passaged using 0.05% Trypsin-EDTA (Thermo Fisher Scientific, 25300054). Cells were monthly tested for mycoplasma contamination using PCR. Cells were routinely FACS sorted for P-RFP+/X-GFP+ to ensure the propagation of a pluripotent population with two X-chromosomes.

**X-GFP reporter**. We used the female F2 ESC line EL16.7 TST, that was derived from a cross of *Mus musculus musculus* with *Mus musculus castaneus*[40]. As a result, cells contain one X chromosome from *M.m musculus* (X$^{mus}$) and one from *M.m castaneus* (X$^{cas}$). Moreover, EL16.7 TST contains a truncation of *Tsix* on X$^{mus}$ (*Tsix*$^{TST/+}$), which abrogates *Tsix* expression and leads to the non-random inactivation of X$^{mus}$ upon differentiation. The EL16.7 TST cell line was obtained from Jeannie Lee, Massachusetts General Hospital (Boston, USA).

A GFP reporter construct was targeted into the second exon of *Hprt* on X$^{mus}$ as follows: Homology arms flanking the target site were amplified from genomic DNA and cloned into pBluescript II SK(+) (Addgene, 212205) by restriction-enzyme based cloning and the cHS4-CAG-nlsGFP-cHS4 construct, kindly provided by J. Nathans[44], was cloned between the two homology arms.

5×10$^6$ cells were mixed with 1.6 µg circularised targeting vector and 5 µg single guide RNA vector PX459 (Addgene, 48139) (5′-TATACCTAATCATTATGCCG-3′), to achieve an optimal ratio of Cas9 to targeting vector equal to 5:1[100]. Cells were nucleofected with the Amaxa Mouse Embryonic Stem Cell Nucleofector Kit (Lonza, VPH-1001) using program A-30 and 7.5 µM RS-1 (Merck, 553510) was added to enhance homology-directed repair. To select for the disruption of *Hprt*, cells were grown in the presence of 10 µM 6-thioguanine (Sigma-Aldrich, A4882-250MG) for 6 days, and GFP+ cells were isolated by FACS using a BD Influx (BD Biosciences). Single clones were screened by Southern blot hybridization. Inactivation of the X-GFP construct upon differentiation was confirmed using embryoid body differentiation.

pSpCas9(BB)-2A-Puro (PX459) was a gift from Feng Zhang (Addgene plasmid # 48139; http://n2t.net/addgene:48139; RRID:Addgene_48139)

**X-chromosome loss assessment using a X dual color reporter**. The EL16.7 clone has been originally selected for its karyotypic stability and absence of X-chromosome loss[39], which has been frequently observed for other female ESC lines[101]. To independently confirm this, we generated a variant of our cell line, in which we simultaneously inserted the X-GFP reporter into the second exon of the *Hprt* locus on the X$^{mus}$, as well as a tdTomato reporter into the same location on the X$^{cas}$ chromosome of EL16.7 TST cells. Integration was performed as described above, using 5×10$^6$ cells with 1.6 µg of each circularised targeting vector and 5 µg single guide RNA vector PX459. Lack of X-chromosome loss was then assessed during prolonged culture in serum plus LIF conditions (described in Materials: Embryonic stem cell culture) by FACS analysis (Supplementary Fig. 1a, b). Indeed, even after 10 passages, 99.5% of cells were X-GFP/X-Tomato double-positive, indicating no significant loss of X chromosomes.

**Reprogramming cassette**. An all-in-one gene targeting vector with doxycycline-inducible reprogramming factors, MKOSimO neotk rtTA Sp3, kindly provided by K. Kaji[43], was targeted into the third intron of the *Sp3* gene in the ESC line EL16.7 TST X-GFP using CRISPR-Cas9. 5×10$^6$ cells were mixed with 3.8 µg circularised targeting vector and 2.5 µg single guide RNA vector PX459 (5′-GTGACAATCT CCGGAAAGCG-3′) and nucleofected with the Amaxa Mouse Embryonic Stem Cell Nucleofector Kit (Lonza, VPH-1001) using program A-24. 7.5 µM RS-1 (Merck, 553510) was added to enhance homology-directed repair. Cells were selected with 300 µg/ml G418 for 5 days. Clones were selected for expression of mOrange upon the addition of 1 mg/ml doxycycline for 24 h and then screened by Southern blot hybridization.

Knockout of mOrange was generated using CRISPR-Cas9. 5×10$^6$ cells were mixed with 1.8 µg single guide RNA vector PX459 V2 (Addgene, 62988) (5′-CAACGAGGACTACACCATCG-3′) and nucleofected with the Amaxa Mouse Embryonic Stem Cell Nucleofector Kit (Lonza, VPH-1001) using program A-30. mOrange-negative cells were isolated by FACS using a BD Influx and single clones were screened for maintenance of proper cassette expression by quantitative RT-PCR. Primers for detection of transcripts by quantitative RT-PCR are listed in Supplementary Table 2.

pSpCas9(BB)-2A-Puro (PX459) V2.0 was a gift from Feng Zhang (Addgene plasmid # 62988; http://n2t.net/addgene:62988; RRID:Addgene_62988).

**Southern blot**. Genomic DNA (10 µg) was digested with appropriate restriction enzymes overnight. Subsequently, genomic DNA was separated on a 0.8% agarose gel and transferred to an Amersham Hybond-XL membrane (GE Healthcare, RPN303S). Probes were synthesized by PCR amplification and labelled with dCTP, [α-32P] (Perkin Elmer, NEG513H250UC) using High Prime (Roche, 11585592001) and hybridization performed in Church buffer.

**P-RFP pluripotency reporter**. Lentivirus, encoding the mouse *Nanog* promoter driving RFP expression, was purchased from System Biosciences (SR10044VA-1). EL16.7 TST X-GFP MKOS ESCs were infected at an MOI of 30 and RFP-positive cells FACS purified using a BD Influx. Single clones were isolated and selected based on proper RFP expression using FACS analysis on a BD LSRFortessa.

**Neural precursor cell differentiation**. ESCs were differentiated to neural precursor cells (NPCs) as described previously[42]. ESCs were seeded at a density of 2.75×10$^5$ cells/cm$^2$ in N2B27 (50% DMEM/F12 (Thermo Fisher Scientific, 21041025), 50% Neurobasal medium (Thermo Fisher Scientific, 12348017), 1x N2 (Thermo Fisher Scientific, 17502048), 1x B27 (Thermo Fisher Scientific, 12587001)) supplemented with 0.4 µM PD0325901 (Selleck Chemicals, S1036-5mg), 3 µM CHIR99021 (Sigma-Aldrich, SML1046-5MG) and 1000 U/ml LIF (ORF Genetics, 01-A1140-0100). 24 h later, cells were dissociated using Accutase (Thermo Fisher Scientific, 00-4555-56) and plated at 2.95×10$^4$ cells/cm$^2$ in RHB-A (Takara Bio, Y40001) on 0.2% gelatin-coated T75 flasks, changing media every other day. On days 6 and 8, media was supplemented with 10 ng/ml EGF (R&D

Systems, 236-EG-200) and 10 ng/ml bFGF (Thermo Fisher Scientific, 13256029) and additionally with 10 μM ROCK inhibitor (Sellekchem, S1049) on day 8. On day 9, cells were dissociated using Accutase (Thermo Fisher Scientific, 00-4555-56) and SSEA1 expressing cells were removed by MACS sorting using Anti-SSEA-1 (CD15) MicroBeads (Miltenyi Biotech, 130-094-530). To completely remove cells that hadn't undergone XCI, cells were stained with SSEA-1 eFluor 660 (Thermo Fisher Scientific, 50-8813-42) for 15 min at 4 °C, washed once with 0.5% BSA in PBS and then SSEA1–/P-RFP–/X-GFP– cells were FACS purified using a BD FACSAria II SORP or a BD Influx (BD Biosciences) at a maximum flow rate of 4,000 ev/s to improve survival. FACS sorted cells were seeded at $3.5×10^5$ cells/cm$^2$ on 0.2% gelatin-coated dishes in RHB-A, supplemented with EGF, FGF, and ROCKi. The medium was changed daily until day 12 when cells reached 100% confluency.

**Reprogramming of neural precursor cells.** Reprogramming of day 12 neural precursor cells was induced by the addition of 1 μg/ml doxycycline (Tocris, 4090/50) and 25 mg/ml L-ascorbic acid (Sigma-Aldrich, A7506-25G) to the NPC medium (RHB-A supplemented with EGF and FGF). 24 h later, cells were dissociated using Accutase and seeded on irradiated mouse embryonic fibroblasts (iMEF) in ESC medium containing 15% FBS and supplemented with 1 μg/ml doxycycline and 25 mg/ml L-ascorbic acid. The medium was changed every other day. To isolate iPSC, SSEA1+/P-RFP+/X-GFP+ cells were isolated using FACS at day 10 of reprogramming and re-plated on 0.2% gelatin-coated plates in ESC medium and kept in doxycycline free conditions for 5–6 days.

**RNA isolation, quantitative RT-PCR and RNA-sequencing.** RNA was extracted using the RNeasy Plus Mini Kit (Qiagen, 74136) or RNeasy Micro Kit (Qiagen, 74004) and quantified by Nanodrop. cDNA was produced with a High-Capacity RNA-to-cDNA Kit (Thermo Fisher Scientific, 4387406) and was used for qRT-PCR analysis in triplicate reactions with Power SYBR Green PCR Master Mix (Thermo Fisher Scientific, 4367659). Libraries were prepared using the TruSeq Stranded Total RNA Library Preparation Kit (Illumina, 20020597) followed by paired-end sequencing (2 ×125 bp) on an Illumina HiSeq 2500.

**Assay for transposase-accessible chromatin with high throughput sequencing (ATAC-seq).** ATAC-seq was performed as described previously[102] with minor modifications. 50,000 FACS purified cells were resuspended in 50 μl cold lysis buffer (10 mM Tris-HCl pH 7.4, 10 mM NaCl, 3 mM MgCl$_2$, 0.01% Digitonin, 0.1% Tween-20, 0.1% IGEPAL CA-630). After 3 min the lysis was washed out using 1 ml cold lysis buffer containing Tween-20, but no Digitonin or IGEPAL CA-630. Cells were centrifuged for 10 min at 500 rcf and 4 °C, supernatant was removed and nuclei were resuspended in 50 μl transposition reaction mix (25 μl Tn5 Transposase buffer (Illumina, 15027866), 2.5 μl Tn5 transposase (Illumina, 15027865), 16.5 μl PBS, 0.01% Digitonin, 0.1% Tween-20, 5 μl nuclease-free water) and incubated at 37 °C for 45 min with 1000 RPM mixing. DNA was isolated using the MinElute PCR Purification Kit (Qiagen, 28004). Library amplification was performed by two sequential PCR reactions (8 and 4–7 cycles, respectively) using the NEBNext High Fidelity PCR Master Mix (New England Biolabs, M0541S). DNA was then double-size selected using 0.5x and 1.5x Agencourt AMPure XP beads (Beckman, A63880) to isolate fragments between 100 bp and 1 kb. Library quality was assessed on a Bioanalyzer, followed by paired-end sequencing (2 ×125 bp) on an Illumina HiSeq 2500.

**Cell isolation and purification for ATAC-seq and RNA-seq.** Cells were dissociated using Accutase (Thermo Fisher Scientific, 00-4555-56) (for NPCs), 0.05% Trypsin-EDTA (Thermo Fisher Scientific, 25300054) (for ESCs) or 0.25% Trypsin-EDTA (Thermo Fisher Scientific, 25200056) (for iPSCs) and then stained with SSEA-1 eFluor 660 (Invitrogen, 50-8813-42, eBioscience eBioMC-480 (MC-480)) at a dilution of 1/100 for 15 min at 4 °C. Cells were washed once with 0.5% BSA in PBS and then FACS sorted using a BD FACSAria II SORP or a BD Influx. Gating strategy is exemplified in Supplementary Fig. 7.

**Cell isolation and purification for Hi-C.** Purification of $G_0G_1$ cells based on DNA content was performed as described previously[54] with minor modifications. Briefly, cells were dissociated using Accutase (Thermo Fisher Scientific, 00-4555-56) (for NPCs), 0.05% Trypsin-EDTA (Thermo Fisher Scientific, 25300054) (for ESCs) or 0.25% Trypsin-EDTA (Thermo Fisher Scientific, 25200056) (for iPSCs) and then stained with SSEA-1 eFluor 660 (Invitrogen, 50-8813-42, eBioscience eBioMC-480 (MC-480)) at a dilution of 1/100. iPSCs were additionally MACS sorted using Anti-SSEA-1 (CD15) MicroBeads (Miltenyi Biotech, 130-094-530). Cells were then fixed for 10 min at room temperature with freshly prepared 1% formaldehyde in PBS (Sigma-Aldrich, F8775-4X25ML) and the reaction then quenched by addition of 0.2 M glycine (NZYTech, MB01401). $1×10^6$ cells/ml were permeabilized using 0.1% saponin (Sigma-Aldrich, 47036-50G-F). 10 μg/ml DAPI (Thermo Fisher Scientific, D1306) and 100 μg/ml RNase A (Thermo Fisher Scientific, EN0531) were added and samples incubated for 30 min at room temperature protected from light with slight agitation. After washing once with cold PBS, samples were resuspended in cold 0.5% BSA in PBS at a concentration of $1×10^7$ cells/ml and immediately FACS purified using a BD FACSAria II SORP or a BD Influx. Gating strategy is

exemplified in Supplementary Fig. 7. After FACS sorting, dry cell pellets were snap-frozen in dry ice and stored at −80 °C.

**In situ Hi-C library preparation.** In situ Hi-C was performed as described previously[34] with minor modifications. One million cells purified for $G_0G_1$ were used as starting materials. Cells were lysed using 250 μl cold lysis buffer (10 mM Tris-HCl pH 8, 10 mM NaCl, 0.2% IGEPAL CA-630) supplemented with 50 μl protease inhibitor cocktail (Sigma-Aldrich, P8340-1ML). Cells were digested with 100 U MboI (New England Biolabs) and incubated for 2 h at 37 °C under rotation, followed by the addition of another 100U for 2 h and another 100U before over-night incubation. The next day a final 100U were added and incubated for 3 h. After fill-in with biotin-14-dATP (Thermo Fisher Scientific, 19524016), ligation was performed with 10,000 U T4 DNA Ligase (New England Biolabs, M0202M) overnight at 24 °C under rotation. After de-crosslinking, DNA was purified using ethanol precipitation and sonicated to an average size of 300–700 bp with a Bioruptor Pico (Diagenode; seven cycles of 20 s on and 60 s off). Ligation products containing biotin-14-dATP were pulled-down using Dynabeads MyOne Streptavidin T1 beads (Thermo Fisher Scientific, 65601) and end-repaired and A-tailed using the NEBNext End Repair/dA-Tailing Module (New England Biolabs, E6060S and E6053S). Libraries were amplified using the NEBNext High Fidelity PCR Master Mix and NEBNext Multiplex Oligos for Illumina (New England Biolabs, M0541S and E7335S) for 8 cycles and size-selected with 0.9x Agencourt AMPure XP beads. Library quality was assessed on a Bioanalyzer and by low-coverage sequencing on an Illumina NextSeq 500, followed by high-coverage paired-end sequencing (2 ×125 bp) on an Illumina HiSeq 2500.

**RNA-fluorescent in situ hybridization.** Strand-specific RNA FISH was performed with fluorescently labelled oligonucleotides (IDT) as described previously[103]. Briefly, cells were fixed with 4% paraformaldehyde for 10 min at room temperature and then permeabilized for 5 min on ice in 0.5% Triton-X. 10 ng/ml equimolar amounts of Cy5 labelled Xist probes BD384-Xist-Cy5-3'-AM (5'-ATG ACT CTG GAA GTC AGT ATG GAG /3Cy5Sp/ -3') and BD417-5'Cy5-Xist-Cy5-3'-AM (5'-/5Cy5/ATG GGC ACT GCA TTT TAG CAA TA /3Cy5Sp/ -3') were hybridized in 40% formamide, 10% dextran sulfate, 2xSSC pH 7 at room temperature overnight. Slides were then washed in 30% formamide 2xSSC pH 7 at room temperature, followed by washes in 2xSSC pH 7 and then mounted with Vectashield (Vector Laboratories, H1200). Images were acquired using an EVOS and a Cy5 light cube (Thermo Fisher Scientific).

**LNA GapmeR Transfection of NPCs.** LNA GapmeRs with the following sequences were ordered from IDT: Control, AACACGTCTATACGC; Xist, TCTTGGTTACTAACAG[104] Jpx, GGACGCCGCCATTTTA[82]. NPCs were differentiated for 9 days as described above, dissociated using Accutase (Thermo Fisher Scientific, 00-4555-56) and SSEA1 expressing cells were removed by MACS sorting using Anti-SSEA-1 (CD15) MicroBeads (Miltenyi Biotech, 130-094-530). $5×10^6$ NPCs were nucleofected with LNA GapmeRs at a final concentration of 5 μM using a Lonza 4D-Nucleofector and the P3 Primary Cell 4D-Nucleofector X Kit (Lonza, V4XP-3024) using program DS-113 according to manufacturer's instructions. RNA was extracted 24 h post-nucleofection using the RNeasy Plus Mini Kit (Qiagen, 74136). Primers for detection of transcripts by quantitative RT-PCR are listed in Supplementary Table 2.

**Allele-specific analysis.** Reads from the PaX hybrid cell line were disambiguated and mapped in two steps. First, each read was mapped in the reference genome of 129S1/SvImJ, and independently in the reference genome of CAST/EiJ (both genomes are available from the Wellcome Trust Sanger Institute ftp://ftp-mouse.sanger.ac.uk/REL-1504-Assembly/) using BWA-MEM version 0.7.17 (with options -L 500 for ATAC-seq and -t4 -P -k17 -U0 -L0,0 -T25 for Hi-C). In each case, the resulting SAM files were processed by samtools version 1.8 with the fixmate option to fill in mate coordinates. Each read was then assigned to a single genome and the coordinates were lifted over to the reference mm10 (Mus musculus genome of strain C57BL/6 J) with custom Python scripts available for download at http://github.com/gui11aume/asmap. Briefly, genome assignment was carried out as follows: if the mapping quality of the read was 0 in each genome, the read was considered unmapped and no assignment was performed; otherwise, the read was assigned to the genome with the best alignment score (field AS:i from the output of BWA-MEM). When both alignment scores were equal, the genome of origin was considered ambiguous and no assignment was performed (but the coordinates of the read were still lifted over to mm10). Lift over to mm10 was also performed with custom Python scripts using the positions of the SNPs and indels of 129S1/SvImJ and CAST/EiJ relative to C57BL/6 J[89]. The positions of the SNPs and indels relative to the reference strain C57BL/6J (mm10) were downloaded from ftp://ftp-mouse.sanger.ac.uk/REL-1505-SNPs_Indels/strain_specific_vcfs, and processed as explained in[89].

**RNA-seq analysis.** Mapping and disambiguation were performed as explained in the section Allele-Specific Analysis. Reads were mapped with STAR[105] (standard options) and the Ensembl mouse genome annotation (GRCm38.78). Gene expression was quantified with STAR (--quantMode GeneCounts). Batch effects

were removed using the ComBat function from the sva R package (v.3.22). Sample scaling and statistical analysis were performed with the R package DESeq2[106] (R v.3.3.2 and Bioconductor v.3.0), and the $\log_2$(vsd) (variance stabilized DESeq2) counts were used for further analysis unless stated otherwise. Standard TPM (transcripts per million) values were used as an absolute measure of gene expression.

**ATAC-seq analysis**. Mapping and disambiguation were performed as explained in the section Allele-Specific Analysis. Peak calling was performed using Zerone[107], with option "make atac" at compile time. The ATAC-seq profiles were discretized using default parameters against the baseline set by the NPC profile. In the process, the four replicates of each time point were merged into a single discretized profile of resolution 300 bp. The NPC profile was discretized separately, with option "--no-mock" to indicate the absence of a separate baseline profile. In all cases, only the windows with a confidence score above 99.9% were considered to be ATAC-seq peaks.

**In situ Hi-C data processing and normalization**. Mapping and disambiguation were performed as explained in the section Allele-Specific Analysis, and only the read pairs where at least one end was mapped unequivocally were kept for further analyses (i.e., with mapping quality greater than 0 and with a non-ambiguous chromosome of origin). Quality of the reads was checked using FastQC (http://www.bioinformatics.babraham.ac.uk/projects/fastqc/).

We used all unambiguous reads to compute raw contact matrices for any given resolution. We then excluded the main diagonal from the analysis, as most of the artefactual reads (self-circles, dangling-ends, duplicated reads, random breaks) appear as contacts with shorter ranges than the corresponding resolution. After that, we excluded bins that presented either low mappability (below 0.5), no restriction enzyme sites, or very low counts. The latter were defined matrix-wise and heuristically, based on the total number of counts per bin (row sums) using the R package dryhic and the function first_max, which computes the lowest maximum value of the density estimate of the input numeric vector.

Finally, we modelled the genomic biases using the R package OneD[108], applying the standard features (mappability, number of restriction enzyme sites, GC content) and the SNP density. This yields the corrected (a.k.a. normalized) matrices used in the downstream analysis. Supplementary Table 1 shows the number of normalized allelic reads of each in situ Hi-C dataset generated in this study.

**Allele-specific expression**. From a list of 806 protein-coding genes on the X-chromosome, we masked genes with insufficient sequence polymorphisms, leaving 558 genes. Moreover, to remove lowly expressed genes, we removed genes where expression from cas was below the 25th percentile, leaving 335 genes that passed these criteria. The allelic expression ratio of these genes was then calculated by dividing mus reads by the sum of mus and cas reads ((mus/mus+cas)). To correct for biases introduced by SNP density variations, the absolute allele-specific expression for mus and cas alleles was calculated by multiplying the bulk counts by the allelic ratio. Furthermore, for analysis involving X-reactivation, only genes biallelically expressed in iPSC and ESC were considered (allelic expression ratio >0.4 and <0.6), leaving 275 genes.

**Identification of A and B compartments**. Two Hi-C sub-matrices were extracted from normalized Hi-C matrices at 100-kb resolution with a split point at *Dxz4* (at coordinates 75.6 Mb). The A/B compartment scores were then computed separately with a standard principal-component approach. Namely, the matrix entries were normalized by the distance decay from the diagonal (i.e., interaction scores were divided by the average interaction score at the given linear distance) and then transformed into correlation matrices using the Pearson product-moment correlation. The first principal component of a PCA (PC1) on each of these matrices was used as a quantitative measure of compartmentalization, and AT-content was used to assign negative and positive PC1 categories to the correct compartments. When necessary, the sign of the PC1 (which is randomly assigned) was inverted so that positive PC1 values corresponded to the A compartment, and vice versa for the B compartment.

To more accurately define A/B-compartments, we then applied a gaussian mixture model with two components ($k = 2$) to the values of the PC1 using the R package mclust[109] using an unequal variance model ("V"). To reduce the impact of extreme outliers we used the 5% trimmed mean of values. The PC1 values were then centred at the intersection of the two gaussian models, separately for each genotype and sample. After centring, values were then normalized to +1 to −1.

To calculate the number of compartments, we separately merged bins of A and B compartments, positive and negative PC1 values respectively, using bedtools "merge"[110] and excluded bins smaller than 300 kb.

**Clustering of A/B compartments**. Clustering was performed utilizing UMAP (uniform manifold approximation and projection) using the R package uwot with the settings n_neighbors = 8, metric = "euclidean" and min_dist = 0.0001.

**Identification of spatial clusters and subcompartments**. Identification of spatial clusters and subcompartments was performed as described previously[59]. To smooth the diagonal decay and other biases, observed-over-expected normalization was applied individually to each contact matrix[4], followed by ICE row-sum balancing[111]. Outliers, such as enhancer-promoter loops, were clipped to the 90-th percentile value of each matrix. On the balanced matrix, the correlation matrix was computed and its diagonal was set to 0. The outliers of the correlation matrix were further smoothed with linear scaling; values below the 5-th and above the 95-th percentiles were clipped to −1 and +1, respectively, and the values within this range were scaled proportionally.

Spatial clusters were identified on the correlation matrix running *k*-means (with 10 restarts) on $k = 12$ weighted eigenvectors, i.e., the 12 leading eigenvectors, each weighted by its respective eigenvalue. Clustering of rows was only performed on D5 P-RFP+ X$^{mus}$. Subcompartment names A1, A2, AB, B1, and B2 were assigned based on their mutual interaction patterns and their PC1 values in NPC X$^{mus}$. For other time points, cluster labels were aligned with the ones found in D5 state.

**Insulation, TAD, and TAD boundary calling**. Normalized contact matrices at 50-kb resolution were used to obtain the insulation score based on the number of contacts between bins on each side of a given bin, using a previously described method[86] with default parameters. To account for variation in SNP density, the signal of the insulation score was further smoothed by using moving averages of span 7 and selecting local minima as TAD borders. A cut-off value of –0.086 was chosen, as it gave a similar number of TADs on the active X as previously reported[5], i.e., a TAD border was inserted at every local minimum with a smoothed insulation score below the cut-off.

**Domain score**. The domain score was used to quantify the degree of connectivity within TADs and was calculated as described previously[53]. Briefly, normalized contact matrices at 50-kb resolution were used. Then for each TAD, the fraction of intra-TAD contacts over its total number of *cis* contacts were used to calculate the domain score.

**Inter- and intracompartment strength measurements**. We followed a previously reported strategy to measure overall interaction strengths within and between A and B compartments[97]. Briefly, we based our analysis on the 100-kb bins showing the most extreme PC1 values, discretizing them by percentiles and taking the bottom 20% as the B compartment and the top 20% as the A compartment. We classified each bin in the genome according to PC1 percentiles and gathered contacts between each category, computing the $\log_2$ enrichment over the expected counts by distance decay. Finally, we summarized each type of interaction (A–A, B–B and A–B/B–A) by taking the median values of the $\log_2$ contact enrichment.

**Inter-mega-domain interactions**. To quantify the degree of inter-mega-domain interactions of subcompartments, we filtered depth-corrected Hi-C contact maps at 50-kb resolution for long-range chromatin interactions (>8.2 Mb) and then selected for strong interactions (top 20%). We then counted the number of strong long-range inter-mega-domain interactions and of all long-range inter-mega-domain interactions for each spatial cluster and then calculated the fraction of strong versus all long-range inter-mega-domain interactions.

**Integration of published ChIP-seq, CHART-seq and DamID data**. Datasets generated by[8] were downloaded from GEO. BigWig files were converted to wig using bigWigToWig and then to bed using wig2bed[112]. Liftover of bed files from mm9 to mm10 was performed using CrossMap[113]. Bedtools "map" and "intersect" functions were then used to integrate data with spatial clusters[110].

**Correlation of subcompartments with repeat elements**. Genomic coordinates of repeat elements (SINE and LINE1) were downloaded from the UCSC Table Browser from the remasker table ("rmsk") for the mm10 reference genome assembly (https://genome.ucsc.edu/cgi-bin/hgTables).

**Visualization**. All plots with the following exceptions were generated using ggplot2[114]. ATAC-seq profiles and peaks, A/B compartments, and insulation score data were plotted using pyGenometracks[115]. ATAC-seq profiles for Fig. 5a were generated using GVIZ[116]. FACS plots were generated using FlowJo. ForceAtlas2 network of spatial cluster interactions was visualized using Gephi[67].

**Statistics and reproducibility**. RNA-seq, ATAC-seq and in situ Hi-C data throughout the paper were generated by analysis of two biologically independent samples. Representative data are shown only if results were similar for both biologically independent replicates. LNA knockdown experiments were performed in four biologically independent replicates. All box plots depict the first and third quartiles as the lower and upper bounds of the box, with a band inside the box showing the median value and whiskers representing 1.5x the interquartile range.

**Reporting summary**. Further information on research design is available in the Nature Research Reporting Summary linked to this article.

## Data availability

The raw and processed data produced in this study are available on Gene Expression Omnibus "GSE157448". We used the following publicly available datasets: "GSE99991" and "GSE116413". All other relevant data supporting the key findings of this study are available within the article and its Supplementary Information files or from the corresponding author upon reasonable request. The source data underlying Fig. 5h is provided as a Source Data file. A reporting summary for this Article is available as a Supplementary Information file. Source data are provided with this paper.

## Code availability

The code used for allele-specific mapping is available for download from GitHub at https://github.com/gui11aume/asmap[117].

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

## Acknowledgements

We thank J. Lee for EL16.7 TST ES cells; K. Kaji for reprogramming cassette plasmids; J. Nathans for X-GFP plasmids; M. Ebisuya for help with LNA nucleofections; R. Stadhouders and G. Stik for advice on Hi-C technology; P. Soler for advice on bioinformatic

analyses; the CRG Genomics Unit for sequencing; the CRG/UPF FACS Unit for FACS sorting; and members of B.P.'s laboratory for discussions. We also thank R. Stadhouders and J. Valcarcel for critical reading of the manuscript. This work was supported by the European Research Council under the 7th Framework Programme FP7/2007-2013 (ERC Synergy Grant 4D-Genome, grant agreement 609989 to G.J.F.), by the Spanish Ministry of Science, Innovation and Universities (BFU2014-55275-P, BFU2017-88407-P to B.P. and PGC2018-099807-B-I00 to G.J.F.), the Agencia Estatal de Investigación (AEI) (EUR2019-103817 to B.P.), the AXA Research Fund (to B.P.) and the Agencia de Gestio d'Ajuts Universitaris i de Recerca (AGAUR, 2017 SGR 346 to B.P.) and by the NIH grant R35GM124926 to S.F.P. We would like to thank the Spanish Ministry of Economy, Industry and Competitiveness (MEIC) to the EMBL partnership and to the "Centro de Excelencia Severo Ochoa". We also acknowledge support of the CERCA Programme of the Generalitat de Catalunya. M.B. was supported by a La Caixa International PhD Fellowship.

## Author contributions

M.B. and B.P. conceived the study and wrote the manuscript with input from all coauthors; M.B. established reporter cell line, performed reprogramming experiments and performed molecular biology, RNA-seq, ATAC-seq, and in situ Hi-C experiments; N.Ü. performed X-loss experiments; M.B., E.V., E.Z., and G.J.F. performed bioinformatic analyses with input from S.F.P.; M.B. and E.V. integrated and visualized data; B.P. and G.J.F. acquired funding and supervised the research.

## Competing interests

The authors declare no competing interests.
