## [Peer Review File · Nature Communications]

REVIEWER COMMENTS

Reviewer #1 (Remarks to the Author):

The authors have developed an elegant approach to investigate X-chromosome reactivation during iPSC reprogramming of neural precursor cells (NPCs), utilizing FACS-sorting via two fluorescent reporters. By integrating ATAC-seq, RNA-seq and Hi-C data from numerous timepoints during iPSC reprogramming, the authors aimed to investigate how 3D-structure interplays with chromatin opening and transcriptional reactivation on the inactive X chromosome.

The authors show that the inactive X in NPCs forms somewhat attenuated A/B-like compartments below its known megadomain structure. The authors then identify several subcompartments that differ from each other in their kinetics during reactivation, gene density, Xist coating and chromatin state. One subcompartment (A1) harbors a large number of escape genes, and a subset of genes in close proximity to these escapees are re-activated early during reprogramming. Finally, they show that establishment of TAD's during X chromosome reactivation (XCR) is initiated prior to gene reactivation.

The manuscript is well written and describes a series of interesting observations. However, the novelty of the results is overstated in some instances, in particular with respect to the finding that A/B compartments exist on the Xi (see detailed comment below). Overall, we feel that the PaX iPSC reprogramming system enables the authors to examine XCR with a higher resolution than previous studies, which allows the authors to verify and complement previous findings. The work is thus a valuable, albeit not ground-breaking resource for the X inactivation and 3D-genome fields.

Major comments:

1. At the end of the first results section (p.6), the authors conclude that X-reactivation “occurs in a switch-like synchronous fashion”. However, the methodology used does not allow this conclusion. By sorting the cells on the X-GFP reporter, the authors select cells that have already undergone XCR, because they have reactivated an X-linked GFP transgene. Since this subset of cells might have initiated XCR gradually over several days before reactivating GFP, the conclusion that XCI occurs switch-like and synchronous is not supported.

2. Along the same lines, the presentation of the data from the different sorted populations is often misleading, in particular when presented as a line plot (Fig. 1e/f, Fig. 4c/e, Fig. 5b/d, Fig. 6h/e, Fig. S1e, Fig. S4b/c/h/i). This representation suggests a temporal order, although different sorted populations are compared. It should be clear to the reader that the different conditions don't represent a continuous progression, but a targeted enrichment of a subpopulation for example as a grouped boxplot, where color indicates the sorting strategy and the X-axis labels the day.

3. The authors suggest that Xist RNA prevents restructuring of the A/B compartments on the Xi (line 211-213), because compartments are largely unchanged at D5 when Xist is still expressed. However, it was not tested whether after Xist down-regulation the compartments had disappeared. Such a conclusion would require additional data directly after X-reactivation (D6 X-GFP+) or at an intermediate state (D6 X-GFP~).

4. The novelty of the observation that the inactive X contains A/B-like compartments is overstated. In the introduction it should be made clear that some previous studies had indeed concluded an absence of A/B compartments in NPCs, but that others had observed them in certain mouse and human tissues (Darrow 2016).

5. Is it correct that two biological replicates were analyzed in each analysis? This should be stated more clearly in the legends and methods section. Moreover, the replicates should be used to test whether the conclusions are reproducible. Instead it seems as if they were usually merged or averaged. For example, we would suggest to show the individual replicates in figure 1c. Moreover, for all main conclusions, it should be shown that they are reproduced in individual replicates.

6. Several major conclusions of the paper are not supported by a statistical analysis of the data. Specifically, statistics should be reported for the data in plots Fig. 3 e/f/g/h/j, Fig 6e, Sup. Fig. 3b/c/d/e/g. In particular, Sup. Figures 3d and e do not seem to support the conclusions in the text (CTCF and RAD21 are depleted in Xist-rich subcompartments). Proper statistics would be needed for this assessment.

7. The analysis in Figure 6 compares TADs that arise early and late, using a relative domain score. A lower relative domain score could mean either that the domain has not yet formed (late TAD) or that it was already present at the initial time point (super-early TAD). Fig. S6d shows that already in NPCs the domain score varies significantly between and within subcompartments. A stratification of TADs based on their domain score in NPCs might help to clarify this issue.

8. To interpret the analyses in Fig. 6 it would be useful to know how the early and late TADs are distributed across the subcompartments. It would thus be helpful to label the bars in Fig. S6f with the absolute TAD numbers.

9. The authors conclude that down-regulation of Jpx precedes the down-regulation of Xist. To allow this comparison the two genes should be compared in the same plot.

10. In Fig. 4f, the number of genes in each cluster should be indicated.

11. We could not find a description of the analysis in Fig. 4g and S4g in the methods section. How were the distances exactly calculated, using the TSS or the gene body? And what annotation for SINE and LINE elements was used?

12. From the fact that at D5, megadomains are visible and TADs start to appear the authors conclude that they are present together. We do not think this conclusion can be drawn given that they might present in different cell subsets in the population.

13. Code used for the analysis should be made available to the reviewers before acceptance for publication.

Additional Comments:

14. The labeling on some figures is very small and therefore hard to see (e.g. legend in Fig. 1c, clusters in Fig. 3b, X-axis in Fig. 4a, axes-text in Fig. 5b/d).

15. Significance in Sup. Fig. 4h is shown using stars, while absolute values are shown in the rest of the plots

16. Figure 5b is labeled with a capital letter

17. The acronyms P-RFP and X-GFP are sometimes complicated to read, when combined. The authors might consider to shorten the acronyms, e.g. to PR and XG or P and X.

18. Reference to Fig. 1b in line 114 should maybe point to Fig. 1a instead?

19. We would suggest to show all cells in Fig. S1b, not only the RFP gates cells.

20. The GFP~ symbol looks very similar to GFP-, we would suggest to change it.

21. In line 269 the average domain size is indicated as 317 kb. Looking at Fig. 3a it seems that this number should maybe multiplied by 50kb?

22. The dots in Fig. 4a are very faint and difficult to see.

23. We would suggest to increase the size of the dots in Fig. 5b+d, because they are difficult to see.

Reviewer #2 (Remarks to the Author):

The manuscript by Bauer et al. describes the process of mouse X chromosome reactivation during a timecourse of reprogramming at the genomic level. This manuscript has made several important findings that have been enabled by their new PaX genetic system which allows them to perform inducible reprogramming, to monitor pluripotency and X reactivation. This system allowed them to purify cells during the timecourse of reprogramming that had or had not acquired Nanog expression, and, had or had not reactivated Hprt from the inactive X chromosome, rather than analyzing bulk cells at different days of reprogramming as had been performed previously. Combined with allele-specific genomic analyses, this has afforded them a powerful system to assess how the inactive X's architecture, accessibility and expression changes during reprogramming.

There are several important insights gained, including: identification of compartments on the mouse inactive X, revealed by their high resolution HiC data; interactions across the hinge of the inactive X; reactivation occurring first at genes in an A sub-compartment near to escapee genes, preceded by increased accessibility at these genes; changes in TAD structure and compartments occurring apparently independently of gene expression changes; and finally, that Xist expression seems to be the first critical and rate limiting step in reprogramming.

I really enjoyed reading this manuscript and I believe it will make an important contribution to the field. There are three areas that require explanation and further experimental support.

Major items

XO cells

It is well known in the field of X inactivation that ES cells and iPSC cells in mouse notoriously lose an X chromosome to become XO in favour of retaining both X chromosomes. The authors don't mention in the results or in the methods how this has been assessed in their ESC or iPSC cultures and any measures taken to exclude XO cells. Can the authors provide explanation for how they removed XO cells, and provide additional experimental validation data and methodological details? This is critical given their claims of compartments found on the Xi, as this could in theory be due to XO populations of cells, which only have Xa (with compartments). If the ESC from which the NPCs were derived had XO cells then these would also be found in the NPCs unless removed.

Related, is this why ESC were used rather than iPSC for the HiC? Were they perhaps less XO than the iPSC? Since the focus of the paper is on reprogramming ESC over iPSC or both, seems a surprising choice.

Heterogeneity in cell populations

The new PaX system certainly allows for much greater purity of cells in different states along the reprogramming timecourse than ever before. However, there must still be heterogeneity at each stage needs to be acknowledged and discussed. All the experiments were bulk genomics experiments. It is therefore unclear what partial changes mean at the mid-reprogramming timepoints, for example, a change in chromatin accessibility to 25% of iPSC levels (Fig 4c), an intermediate state from HiC data. Does the 25% change in accessibility mean 25% of cells have made the switch and the others haven't? Given that the RNAseq, ATACseq and HiC datasets were not performed in the same cells, and that there is often a mixed signal at these mid-reprogramming times, there needs to be careful and specific acknowledgement and discussion of the limits of this analysis.

Related to this, the authors describe compartments on the Xi, and indeed even new interactions between subcompartments across the Dxz4 hinge region. It is important that new findings are confirmed by other methods, such as DNA FISH. DNA FISH could be used to reinforce the A5-A6 interaction across the hinge region. It would also afford them the opportunity to measure the frequency of these events compared with within megadomain interactions. This would enhance their paper by not only providing independent validation but also enriching discussion of heterogeneity.

Functional analysis

The paper has a wealth of genomics data, which allows them to describe the order of events they observe. The authors also provide interpretation that goes beyond the descriptive. For example, the authors postulate that Xist reduction may be driven by changes in Jpx, and that Xist reduction is the rate-limiting/ driving event in Xi reprogramming. This seems like an eminently testable hypothesis, and their paper would benefit from such an approach, as it would extend it beyond describing the process to testing their findings.

Minor items

Their findings of differential Xist binding and K27me/K9me in A and B compartments is expected and known, because A and B compartments represent gene rich and gene poor regions respectively. This should be more clearly acknowledged and referenced.

It isn't clear why Xist is not shown for the mus and cas X in Figure 5b? It is also unclear why the mus and cas expression of the XIC factors completely overlies in the more pluripotent cells. This seems very odd and needs explanation.

The authors ask if gene activation precedes ATACseq opening, or vice versa. This is done for all early reactivating genes or late. This would be enhanced if there was a correlation made between RNAseq and ATACseq, so each gene is compared for its expression level and ATAC signal. The same would be true for other comparisons – can correlations be added? E.g Supp Figure 4c, Supp Figure 4h/i

Wording/ refs

For Fig 5c-g the timing difference in the two sets of changes isn't well described. It would be better to split out the early change and describe, then the later change here.

D4 SSEA+ is missing from Figure 4d but would be useful so should be added.

Reference Supp Figure 4e is missing.

Sentence on lines 183-185 is unclear and should be split into two.

Line 257. 'In light of' rather than 'In front of'.

Line 289 references to Brockdorff lab papers on the polycomb pathway should be included – Pintacuda et al and Alemeida et al.

Reviewer #3 (Remarks to the Author):

In this manuscript, authors reported that they found A/B-like compartments on the inactive X harbouring multiple subcompartments by combining a tailor-made mouse iPSC-reprogramming system and high-resolution Hi-C to produce the time-course combining gene reactivation, chromatin opening and chromosome topology during X-reactivation. Furthermore, they found that TAD formation precedes transcription. This topic is interesting. However, the following concerns may need to be addressed.

Major concerns

1. Although both ESC and iPSC are pluripotent stem cells, they are not the same cell type. To my knowledge, there is no study showing that the chromosome structure of ESC and iPSC is exactly the same. Therefore, I think the authors should analyze the Hi-C data of iPSC instead of ESC during the NPC reprogramming process.

2. The authors aimed to use a new report model to study the compartments and TAD

structural characteristics of X chromosome reactivation. As a matter of fact, the model is mainly divided into two parts: the inactivation process of X chromosome during ESC differentiation into NPC, and the process of X chromosome reactivation during the process of reprogramming NPC to iPSC. A large number of papers have been published on the X-chromosome structural characteristics of these two processes (Luca et al, nature, 2016; Janiszewski et al, 2019, genome research; Pasque et al, cell, 2014, Cantone et al, Nature communications, 2016). For example, Luca et al found the structural organization of the inactive X chromosome in the mouse by comparing NPC and ESC using Hi-C, ATAC-Seq, and RNA-seq. This manuscript is similar not only in experimental materials but also in research methods with Luca et al. Therefore, the authors' description about "a novel reporter model system" is inappropriate.

3. Please explain why the authors chose D5 P-RFP+ which is closer to NPC at

the RNA-seq or ATAC-seq level, as the intermediate, rather than D6 P-RFP+ or D6 X-GFP- which had obvious changes in RNA-seq. Is it because the positive rate of P-RFP cells is higher on the D5?

4. It is better to do biological experiment to confirm the relationship between Jpx and Xist.

5. There is no information about the Hi-C data in the manuscript. A supplementary of Hi-C experiment datasets with valid interaction read number should be provided.

Minor concerns

1. There are wrong labeled in some figures. Such as “Fig 1C X-GFP~” should be revised as X-GFP-.
2. The format of some references in the manuscript does not meet the requirements of this journal.

Point-by-point response to the reviewers' comments

Author's response: We would like to thank the reviewers for their constructive comments, which guided us in our revision process and helped us to produce a strongly improved manuscript. As you will see from the manuscript and our point-by-point response below, we could address every point raised by the reviewers. We added new experimental data and analysis, as for example we could show that X-chromosome loss (first major point of Reviewer #2) is not a concern with our cell line (Supplementary Figure 1a, b). We also added exciting new functional data (suggested by both Reviewer #2 and Reviewer #3) by our LNA-knockdown experiments (Fig. 5h), in which we showed that knockdown of Jpx RNA is sufficient for the downregulation of Xist RNA, while Jpx RNA is not affected by Xist RNA knockdown. This causal dependency of Xist RNA on Jpx expression and the strikingly similar kinetics of Xist and Jpx RNA downregulation during reprogramming (Fig. 5g) strongly support our hypothesis that Jpx downregulation might be an important driver of Xist downregulation during X-reactivation. Furthermore, we added new analysis in Figure panels 1c, 1d, 2d and Supplementary Figures 2e, 4b, 4k, 6e, 6f and reformatted the figures and text according to the reviewers' comments and to increase readability.

Reviewer #1

The authors have developed an elegant approach to investigate X-chromosome reactivation during iPSC reprogramming of neural precursor cells (NPCs), utilizing FACS-sorting via two fluorescent reporters. By integrating ATAC-seq, RNA-seq and Hi-C data from numerous timepoints during iPSC reprogramming, the authors aimed to investigate how 3D-structure interplays with chromatin opening and transcriptional reactivation on the inactive X chromosome.

The authors show that the inactive X in NPCs forms somewhat attenuated A/B-like compartments below its known megadomain structure. The authors then identify several subcompartments that differ from each other in their kinetics during reactivation, gene density, Xist coating and chromatin state. One subcompartment (A1) harbors a large number of escape genes, and a subset of genes in close proximity to these escapees are re-activated early during reprogramming. Finally, they show that establishment of TAD's during X chromosome reactivation (XCR) is initiated prior to gene reactivation.

The manuscript is well written and describes a series of interesting observations. However, the novelty of the results is overstated in some instances, in particular with respect to the finding that A/B compartments exist on the Xi (see detailed comment below). Overall, we feel that the PaX iPSC reprogramming system enables the authors to examine XCR with a higher resolution than previous studies, which allows the authors to verify and complement previous findings. The work is thus a valuable, albeit not ground-breaking resource for the X inactivation and 3D-genome fields.

Author's Response: We thank the reviewer for the generally positive assessment of our manuscript and for the constructive comments, which helped us in improving our manuscript substantially.

Major comments

1. At the end of the first results section (p.6), the authors conclude that X-reactivation “occurs in a switch-like synchronous fashion”. However, the methodology used does not allow this conclusion. By sorting the cells on the X-GFP reporter, the authors select cells that have already undergone XCR, because they have reactivated an X-linked GFP transgene. Since this subset of cells might have initiated XCR gradually over several days before reactivating GFP, the conclusion that XCI occurs switch-like and synchronous is not supported.

2. Along the same lines, the presentation of the data from the different sorted populations is often misleading, in particular when presented as a line plot (Fig. 1e/f, Fig. 4c/e, Fig. 5b/d, Fig. 6h/e, Fig. S1e, Fig. S4b/c/h/i). This representation suggests a temporal order, although different sorted populations are compared. It should be clear to the reader that the different conditions don't represent a continuous progression, but a targeted enrichment of a subpopulation for example as a grouped boxplot, where color indicates the sorting strategy and the X-axis labels the day.

Author's Response to comments 1 + 2: We agree that we had conflated the stages with time — mostly for readability — but they still represent a pseudo-progression. Since the

majority of early P-RFP+ cells eventually switch to become X-GFP+ (Fig. 1b), sorting acts as a resynchronization; it is not the same thing as a multi-step selection on a population of cells with a stochastic fate. Similar sorting strategies to enrich for populations along the correct reprogramming trajectory and their representation as a temporal progression have been used in multiple previous hallmark studies on iPSC-reprogramming and are standard in the field. For examples see (Polo et al. 2012; Knaupp et al. 2017; Schwarz et al. 2018; Janiszewski et al. 2019). Without sorting, cells would be included, which are refractory to X-reactivation and which branch off from the iPSC-reprogramming trajectory as seen in our new analysis of P-RFP+/X-GFP- cells in the PCA of RNA-Seq data (**Fig. 1c, d** and **Supplementary Fig. 1e**).

We have tried to make the figures less suggestive of a temporal order, but this has unfortunately mostly not been possible because of aesthetic or visual constraints. **We have however highlighted the enrichment step between the stages in the text and toned down our statement on switch-like X-reactivation (page 6) from:**

*“In conclusion, the unique properties of the PaX system revealed that X-reactivation during iPSC reprogramming **occurs in a switch-like synchronous fashion**,”*

to: *“In conclusion, the unique properties of the PaX system revealed that X-reactivation during iPSC reprogramming **occurs rapidly**,”*

Moreover, in our reanalysis (modified Fig. 1c), we observed that D6 P-RFP+/X-GFP- cells actually started to deviate from the reprogramming trajectory we defined previously, which made us decide to remove this subpopulation from any further analysis. We believe this improved the robustness of our findings and among others, clarified the relationship of *Xist* and *Jpx*, as we found that downregulation kinetics of the two became highly similar after the removal of this population. Furthermore, our reprogramming trajectory therefore only includes two populations from the same day, at day 6, addressing the comments made.

3. The authors suggest that *Xist* RNA prevents restructuring of the A/B compartments on the Xi (line 211-213), because compartments are largely unchanged at D5 when *Xist* is still expressed. However, it was not tested whether after *Xist* down-regulation the compartments had disappeared. Such a conclusion would require additional data directly after X-reactivation (D6 X-GFP+) or at an intermediate state (D6 X-GFP~).

Author’s response: The statement is indeed based on correlation and due to its speculative nature it should not have been in the Results section. We are presently not in a position to perform the experiments that would establish causality, **we have therefore rephrased the statement (page 10) from:** *“This suggests that the presence of *Xist* and its associated chromatin state delay changes of the Xi compartment structure until *Xist* becomes fully downregulated during reactivation (Figure 1e).”* **to:** *“This suggests that the repressive chromatin state of the Xi might delay the remodelling of its compartment structure.”*

4. The novelty of the observation that the inactive X contains A/B-like compartments is overstated. In the introduction it should be made clear that some previous studies had indeed concluded an absence of A/B compartments in NPCs, but that others had observed them in certain mouse and human tissues (Darrow 2016).

Author's response: Thank you for pointing this out. Scholarship is important for us and although we have already mentioned this previously in our discussion **we have now additionally given due credits to this study**

in the introduction (page 3): *Spatial proximity maps obtained by Hi-C from neural precursors cells (NPCs)(Giorgetti et al. 2016; Wang et al. 2018), as well as from fibroblasts (Wang et al. 2019; Darrow et al. 2016), have suggested that the mouse Xi lacks compartments, with exception of weak compartmentalization observed in mouse brain cells (Darrow et al. 2016).*

and in the results on (page 9): *“Strikingly, this revealed an underlying compartment structure on the inactive X^{mus} in NPCs (Fig. 2b), as originally suggested by Darrow et al.¹³.*

5. Is it correct that two biological replicates were analyzed in each analysis? This should be stated more clearly in the legends and methods section. Moreover, the replicates should be used to test whether the conclusions are reproducible. Instead it seems as if they were usually merged or averaged. For example, we would suggest to show the individual replicates in figure 1c. Moreover, for all main conclusions, it should be shown that they are reproduced in individual replicates.

Author's response: Yes, for each analysis, two biological replicates were analysed. However, due to lack of space, we have only added this information under methods, statistics. Moreover, the individual replicates are now shown in Fig. 1c, and among others have as well been added to Fig. 2c for the compartment data and Fig. 5g for the Jpx/Xist correlation.

6. Several major conclusions of the paper are not supported by a statistical analysis of the data. Specifically, statistics should be reported for the data in plots Fig. 3 e/f/g/h/j, Fig 6e, Sup. Fig. 3b/c/d/e/g. In particular, Sup. Figures 3d and e do not seem to support the conclusions in the text (CTCF and RAD21 are depleted in Xist-rich subcompartments). Proper statistics would be needed for this assessment.

Author's response: We assume that statistical analysis refers to statistical tests. We included very few p-values in the first version of the manuscript because they draw the attention away from the effect size, especially for large samples (see points 2, 5 and 6 of the 2016 statement of the American Statistical Association on the concerns of reporting statistical significance <https://www.amstat.org/asa/files/pdfs/P-ValueStatement.pdf>). However, we understand that this may be misinterpreted as lack of statistical rigour so **we have added the information about the statistical tests where requested**. Furthermore, this showed that differences in CTCF occupancy are significantly different between subcompartments (Supplementary Fig. 3d), while we found differences for RAD21 to not be significant (Supplementary Fig. 3e).

7. The analysis in Figure 6 compares TADs that arise early and late, using a relative domain score. A lower relative domain score could mean either that the domain has not yet formed (late TAD) or that it was already present at the initial time point (super-early TAD). Fig. S6d shows that already in NPCs the domain score varies significantly between and within

subcompartments. A stratification of TADs based on their domain score in NPCs might help to clarify this issue.

Author's response: We have now performed a similar analysis showing the correlation between the absolute domain score in NPC and the relative domain score at D5, highlighting the subcompartments (**new Supplementary Supplementary Fig. 6e**). The plot shows five TADs of the A1 subcompartment with a low relative score at D5 and a high absolute score in NPC, but otherwise TADs with a relative score at D5 in the range 0.0–0.1 span all the absolute scores in NPC and all subcompartments. Except for those five TADs, the initial domain score in NPC has little influence on the relative score at D5, confirming that the score is a reasonable proxy for the timing of TAD appearance. **We have added a reference to Supplementary Supplementary Fig. 6e in the main text (page 26):** *"Importantly, we did not find that differential domain score dynamics of subcompartments were influenced by differences already present in NPCs, as we observed that the initial domain score in NPCs had little influence on the relative score at D5 (Supplementary Fig. 6e). This confirmed that the domain score is a reasonable proxy for the timing of TAD appearance."*

8. To interpret the analyses in Fig. 6 it would be useful to know how the early and late TADs are distributed across the subcompartments. It would thus be helpful to label the bars in Fig. S6f with the absolute TAD numbers.

Author's response: We have changed the figure accordingly (this panel is now Supplementary Supplementary Fig. 6h).

9. The authors conclude that down-regulation of *Jpx* precedes the down-regulation of *Xist*. To allow this comparison the two genes should be compared in the same plot.

Author's response: Fig. 5g now shows an overlay of *Jpx* and *Xist*. The plot shows that the *Jpx* kinetics are actually similar to those of *Xist*. **The text has been changed accordingly.**

10. In Fig. 4f, the number of genes in each cluster should be indicated.

Author's response: We have changed the figure accordingly.

11. We could not find a description of the analysis in Fig. 4g and S4g in the methods section. How were the distances exactly calculated, using the TSS or the gene body? And what annotation for SINE and LINE elements was used?

Author's response: TSS were used. The legends of Fig. 4f, 4g and Supplementary Fig. 4h have been updated accordingly. LINE and SINE repeats for mm10 were obtained from UCSC. **This information has been added to the methods section (new paragraph named "Correlation of subcompartments with repeat elements").**

12. From the fact that at D5, megadomains are visible and TADs start to appear the authors conclude that they are present together. We do not think this conclusion can be drawn given that they might present in different cell subsets in the population.

Author's response: This is an important point. To test this, we mixed *in silico* the Hi-C maps of NPC and ESC at different ratios and we measured the relative domain scores in the subcompartments (new **Supplementary Supplementary Fig. 6f**). The real domain scores at D5 are not compatible with a single mixing ratio. For instance, the TADs of A1 would suggest that ~15% of the cells are in an ESC-like state, but the TADs of B2 require this number to be ~30%. This shows that the transitory structure at D5 is not the superposition of NPC-like cells (with mega-domains and no TADs) and ESC-like cells (with TADs and no mega-domains).

We also represented the mega-domain strength by plotting the values of the first principal component around the junction (**Fig. 2c**), which has been used previously to quantify mega-domain signals and was shown to be proportional to the fraction of mega-domain positive cells (Froberg et al. 2018). This confirmed that the mega-domains are almost equally strong at D5 as in NPC — as visible on **Fig. 2a**. Therefore, the emergence of TADs at D5 is not at the expense of mega-domains, speaking against the hypothesis that TADs appear in the cells that have lost the mega-domains.

In summary, it is more plausible that TADs and mega-domains coincide in the same cells, but the evidence is indirect. **We have clarified this in the text by adding on page 26 the following:**

“While single-cell structural data would be necessary to directly resolve the temporal relationship of TAD formation and loss of the mega-domain structure, our data strongly suggest that the transitory structure observed at D5 is unlikely the result of a heterogeneous cell population and that therefore, TAD formation occurs in the presence of the two mega-domains.”

13. Code used for the analysis should be made available to the reviewers before acceptance for publication.

Author's response: The code is available on Github (<https://github.com/gui11aume/asmmap>) and follows the Nature guidelines for sharing code in publications, including README files and tests (<https://media.nature.com/full/nature-cms/documents/GuidelinesCodePublication.pdf>). The data has been deposited to GEO (GSE157448) and the embargo will be lifted as soon as the article is accepted for publication. **This information has been added to the methods section in the paragraph “Data availability”.**

Additional Comments

14. The labeling on some figures is very small and therefore hard to see (e.g. legend in Fig. 1c, clusters in Fig. 3b, X-axis in Fig. 4a, axes-text in Fig. 5b/d).

Author's response: We have increased the font sizes in these figures.

15. Significance in Sup. Fig. 4h is shown using stars, while absolute values are shown in the rest of the plots

Author's response: The stars have been removed and replaced by p-values.

16. Figure 5b is labeled with a capital letter

Author's response: This has been fixed.

17. The acronyms P-RFP and X-GFP are sometimes complicated to read, when combined. The authors might consider to shorten the acronyms, e.g. to PR and XG or P and X.

Author's response: With the permission of the Reviewer and the Editor, we would like to keep the acronyms as they were in the first version. We agree that they might appear cumbersome, but we believe them to be easier to understand and X-GFP to follow the nomenclature of previous studies using X-linked reporters.

18. Reference to Fig. 1b in line 114 should maybe point to Fig. 1a instead?

Author's response: This has been fixed.

19. We would suggest to show all cells in Fig. S1b, not only the RFP gates cells.

Author's response: Supplementary Supplementary Fig. 1b (now Supplementary Fig. Supplementary Fig. 1c) has been updated accordingly.

20. The GFP~ symbol looks very similar to GFP-, we would suggest to change it.

Author's response: GFP~ has been replaced by GFPint.

21. In line 269 the average domain size is indicated as 317 kb. Looking at Fig. 3a it seems that this number should maybe multiplied by 50kb?

Author's response: The domain size does not appear on Fig. 3a. The 50 kb bins were rearranged based on their interactions, regardless of whether they were contiguous in the genome. The Reviewer probably refers to the average coverage of the clusters. **We have clarified that the average refers to contiguous domains in the text.**

22. The dots in Fig. 4a are very faint and difficult to see.

Author's response: Dots are now bigger.

23. We would suggest to increase the size of the dots in Fig. 5b+d, because they are difficult to see.

Author's response: This has been done.

Reviewer #2

The manuscript by Bauer et al. describes the process of mouse X chromosome reactivation during a timecourse of reprogramming at the genomic level. This manuscript has made several important findings that have been enabled by their new PaX genetic system which allows them to perform inducible reprogramming, to monitor pluripotency and X reactivation. This system allowed them to purify cells during the timecourse of reprogramming that had or had not acquired Nanog expression, and, had or had not reactivated Hprt from the inactive X chromosome, rather than analyzing bulk cells at different days of reprogramming as had been performed previously. Combined with allele-specific genomic analyses, this has afforded them a powerful system to assess how the inactive X's architecture, accessibility and expression changes during reprogramming.

There are several important insights gained, including: identification of compartments on the mouse inactive X, revealed by their high resolution HiC data; interactions across the hinge of the inactive X; reactivation occurring first at genes in an A sub-compartment near to escapee genes, preceded by increased accessibility at these genes; changes in TAD structure and compartments occurring apparently independently of gene expression changes; and finally, that Xist expression seems to be the first critical and rate limiting step in reprogramming.

I really enjoyed reading this manuscript and I believe it will make an important contribution to the field. There are three areas that require explanation and further experimental support.

Author's Response: We thank the reviewer for the enthusiastic comments and are happy to hear that our manuscript was an enjoyable read. We are also grateful for the constructive comments, which we address hereby below.

Major items

XO cells

It is well known in the field of X inactivation that ES cells and iPSC cells in mouse notoriously lose an X chromosome to become XO in favour of retaining both X chromosomes. The authors don't mention in the results or in the methods how this has been assessed in their ESC or iPSC cultures and any measures taken to exclude XO cells. Can the authors provide explanation for how they removed XO cells, and provide additional experimental validation data and methodological details? This is critical given their claims of compartments found on the Xi, as this could in theory be due to XO populations of cells, which only have Xa (with compartments). If the ESC from which the NPCs were derived had XO cells then these would also be found in the NPCs unless removed.

Author's Response: The parental EL16.7 clone that was used in this study was specifically selected for its previously shown high karyotypic stability, with two X-chromosomes being stably transmitted for over 40 doubling times in culture (Lee and Lu 1999). Moreover, we routinely FACS-sorted ESC used to establish PaX to keep only double-positive cells (clarified now in the methods). Those cells with an X-GFP+/P-RFP+ phenotype have two X chromosomes.

To directly test the stability of our cell line experimentally, we used an Xmus-GFP/Xcas-tomato double reporter cell line of the same parental clone and assessed X-loss over 10 passages in serum+LIF conditions. The results of this new experiment, which were now added to Supplementary Fig. 1 (a, b), showed very high stability of the X, with less than 1% of X-loss (*i.e.*, *i.e.*, loss of either X-GFP or X-tomato expression).

Related, is this why **ESC were used rather than iPSC for the HiC?** Were they perhaps less XO than the iPSC? Since the focus of the paper is on reprogramming ESC over iPSC or both, seems a surprising choice.

Author's Response: The reason for using ESC instead of iPSC was not related to XO cells. It was a technical decision motivated by the work of our colleagues who used ESCs as fully pluripotent cell type for Hi-C comparisons of reprogramming time course experiments (Stadhouders et al. 2018). The experiments were performed with the same protocols, the analyses were done with the same material and sometimes by the same people. Using ESCs allowed us to save a lot of time and resources in the initial phase because we could recycle the know-how and have strong internal controls for the quality of the experiments. Moreover, it has been shown previously that 3D genomes of iPSCs and of ESCs are overall highly similar, as iPSC genomes were found to generally adopt an ESC-like higher-order structure again (Krijger et al. 2016). Therefore our choice of ESCs as fully pluripotent cell type in our Hi-C analysis is justified biologically and experimentally.

Heterogeneity in cell populations

The new PaX system certainly allows for much greater purity of cells in different states along the reprogramming timecourse than ever before. However, there must still be heterogeneity at each stage needs to be acknowledged and discussed. All the experiments were bulk genomics experiments. It is therefore unclear what partial changes mean at the mid-reprogramming timepoints, for example, a change in chromatin accessibility to 25% of iPSC levels (Fig 4c), an intermediate state from HiC data. Does the 25% change in accessibility mean 25% of cells have made the switch and the others haven't? Given that the RNAseq, ATACseq and HiC datasets were not performed in the same cells, and that there is often a mixed signal at these mid-reprogramming times, there needs to be careful and specific acknowledgement and discussion of the limits of this analysis.

Author's Response: This is an important point that requires conceptual clarifications. Intermediate ATAC-seq and Hi-C signals are always due to a mixture of cells because they measure digital events (at a given locus there can be only one insertion of the ATAC-seq adapters or only one ligation per genome). A change of 25% in accessibility means that either 25% of the cells have switched, or that in the transitory regime, cells spend 25% of the time in the "accessible" state. The same goes for Hi-C, but for RNA-seq it is also possible that gene expression goes up by 25% in every cell. The PaX system merely ensures that the cells undergo X-reactivation in a synchronous fashion, but it cannot guarantee that the events are the same in every cell because this depends on the underlying biology. **This has been clarified in the text on page 17 by adding the following sentence:**

"However, while we intended to ensure that the isolated subpopulations are homogeneous, due to the analysis of bulk data, we can not exclude the possibility that our observations are attributable to a full reactivation of these genes in only a subset of cells."

Related to this, the authors describe compartments on the Xi, and indeed even new interactions between subcompartments across the Dxz4 hinge region. It is important that new findings are confirmed by other methods, such as **DNA FISH**. DNA FISH could be used to reinforce the A5-A6 interaction across the hinge region. It would also afford them the opportunity to measure the frequency of these events compared with within megadomain interactions. This would enhance their paper by not only providing independent validation but also enriching discussion of heterogeneity.

Author's Response: We agree that important results have to be cross-checked by different methods. However, DNA FISH will have to be performed in follow-up studies, possibly focused on heterogeneity. Our point of view is that this orthogonal verification is not absolutely essential for the validity of our results, as Hi-C is not less reliable than DNA FISH. Moreover, integration of both types of data has proven to be challenging as contact frequencies were demonstrated to be distinct from average spatial distances, therefore requiring careful design for cross-validation of Hi-C and DNA FISH (Fudenberg and Imakaev 2017; Giorgetti and Heard 2016). Adding those results would bring very significant delays to the publication of this work because such a complex type of analysis would have to be newly established and we are presently working at reduced capacity due to the current measures against the COVID-19 pandemic.

Functional analysis

The paper has a wealth of genomics data, which allows them to describe the order of events they observe. The authors also provide interpretation that goes beyond the descriptive. For example, the authors postulate that Xist reduction may be driven by changes in Jpx, and that Xist reduction is the rate-limiting/ driving event in Xi reprogramming. This seems like an eminently testable hypothesis, and their paper would benefit from such an approach, as it would extend it beyond describing the process to testing their findings.

Author's response: We absolutely agree with the reviewer's point of view, which is also in line with the comment of reviewer 3 below, and therefore have added new experiments directly showing the relationship of *Jpx* and *Xist* already in NPCs, by performing LNA-mediated knockdowns (Fig. 5h). This data shows that knockdown of *Jpx* RNA is sufficient for the downregulation of *Xist* RNA, while *Jpx* RNA is not affected by *Xist* RNA knockdown. This causal dependency of *Xist* RNA on *Jpx* expression and the strikingly similar kinetics of *Xist* and *Jpx* RNA downregulation during reprogramming (Fig. 5g) strongly support our hypothesis that *Jpx* downregulation might be an important driver of *Xist* downregulation during X-reactivation. These exciting new results strongly improve our manuscript and we thank the reviewer for suggesting this experiment.

Minor items

Their findings of differential Xist binding and K27me/K9me in A and B compartments is expected and known, because A and B compartments represent gene rich and gene poor regions respectively. This should be more clearly acknowledged and referenced.

Author's Response: We apologize if our tone lacked scholarship in the previous version. **We have rephrased this section on page 14 by writing:**

“On the contrary, H3K9me2-associated protein CBX1 was enriched in B-like subcompartments (**Fig. 3h**), which was expected considering that H3K27me3 and H3K9me2/3 were shown to occupy distinct subcompartments on autosomes (Giorgetti et al. 2016; Rao et al. 2014; Minajigi et al. 2015; Deng et al. 2015).”

It isn't clear why *Xist* is not shown for the *mus* and *cas X* in Figure 5b? It is also unclear why the *mus* and *cas* expression of the XIC factors completely overlap in the more pluripotent cells. This seems very odd and needs explanation.

Author's Response: *Xist mus* and *cas RNA* expression now appear together in Supplementary Fig. 5b. Regarding the second point, once the repressive environment of the Xi is removed, both alleles are subject to the same transcriptional regulation, therefore showing the same expression. As expected, *Xic* genes show biallelic expression after X-reactivation has occurred and behave therefore like most X-linked genes in pluripotent stem cells.

The authors ask if gene activation precedes ATACseq opening, or vice versa. This is done for all early reactivating genes or late. This would be enhanced if there was a correlation made between RNAseq and ATACseq, so each gene is compared for its expression level and ATAC signal. The same would be true for other comparisons – can correlations be added? E.g Supp Figure 4c, Supp Figure 4h/i

Author's Response: We have added a comparison for all genes analyzed, which shows step changes separately for each gene for chromatin opening and gene expression in Supplementary Fig. 4k.

Wording/ refs

For Fig 5c-g the timing difference in the two sets of changes isn't well described. It would be better to split out the early change and describe, then the later change here.

Author's Response: We thank the reviewer for the suggestion and **have rewritten this section to clarify the timing differences.**

D4 SSEA+ is missing from Figure 4d but would be useful so should be added.

Author's Response: The time point has been added.

Reference Supp Figure 4e is missing.

Author's Response: The reference (now Supplementary Fig. 4f) has been added to the text.

Sentence on lines 183-185 is unclear and should be split into two.

Author's Response: The sentence has been split.

Line 257. 'In light of' rather than 'In front of'.

Author's Response: This has been fixed.

Line 289 references to Brockdorff lab papers on the polycomb pathway should be included – Pintacuda et al and Alemeida et al.

Author's Response: The references have been added.

Reviewer #3

In this manuscript, authors reported that they found A/B-like compartments on the inactive X harbouring multiple subcompartments by combining a tailor-made mouse iPSC-reprogramming system and high-resolution Hi-C to produce the time-course combining gene reactivation, chromatin opening and chromosome topology during X-reactivation. Furthermore, they found that TAD formation precedes transcription. This topic is interesting. However, the following concerns may need to be addressed.

Author's Response: We thank the reviewer for the constructive comments, which we addressed below in order to improve our study.

Major concerns

1. Although both ESC and iPSC are pluripotent stem cells, they are not the same cell type. To my knowledge, there is no study showing that the chromosome structure of ESC and iPSC is exactly the same. Therefore, I think the authors should analyze the Hi-C data of iPSC instead of ESC during the NPC reprogramming process.

Author's Response: We do not claim that the 3D genome structure of ESCs and iPSCs are exactly the same, but it has been shown previously that they are overall highly similar, as iPSC genomes were found to generally adopt an ESC-like higher-order structure again (Krijger et al. 2016). Therefore our choice of ESCs as fully pluripotent cell type in our Hi-C analysis is biologically justified.

Importantly, the focus of our study lies elsewhere: First, we identify compartments on the Xi in NPCs, we then observe the emergence of TADs while the mega-domains are still present at D5, we define five sub-compartments from the Hi-C matrix at D5, and we observe an anti-correlation between *Xist* binding and early TAD formation at D5. The Hi-C map of ESC is merely used for contrast, but the phenomena of interest reported here take place in the early time points between NPCs and D5 of reprogramming.

With our focus being on these early events, we decided to use ESCs instead of iPSCs for technical reasons. Our colleagues had recently set up Hi-C protocols with ESCs which were used as fully pluripotent cell type for Hi-C comparisons of reprogramming time course experiments (Stadhouders et al. 2018), so we decided to recycle the know-how to save time in the early phase of the project — some authors of the present study are co-authors with Stadhouders *et al.* Once we obtained confirmation that the quality of the Hi-C maps was optimal, we kept the Hi-C maps in ESCs as our reference point for fully pluripotent stem

cells. In part because of the very high cost and time required of repeating the experiment in iPSCs, but more importantly, because iPSC data would have given little additional information, as this was not the focus of our study as elaborated above.

2. The authors aimed to use a new reporter model to study the compartments and TAD structural characteristics of X chromosome reactivation. As a matter of fact, the model is mainly divided into two parts: the inactivation process of X chromosome during ESC differentiation into NPC, and the process of X chromosome reactivation during the process of reprogramming NPC to iPSC. A large number of papers have been published on the X-chromosome structural characteristics of these two processes (Luca et al, nature, 2016; Janiszewski et al, 2019, genome research; Pasque et al, cell, 2014, Cantone et al, Nature communications, 2016). For example, Luca et al found the structural organization of the inactive X chromosome in the mouse by comparing NPC and ESC using Hi-C, ATAC-Seq, and RNA-seq. This manuscript is similar not only in experimental materials but also in research methods with Luca et al. Therefore, the authors' description about "a novel reporter model system" is inappropriate.

Author's Response: We thank the reviewer for the comment; **we have substituted the term "novel" with "tailor-made" throughout the manuscript** (e.g., in the first title of the results and in the title of Fig. 1) because the individual parts of the PaX system could indeed be perceived as not entirely novel when seen in isolation.

Nevertheless, what makes our PaX system so special is the unique combination of all its features as described in Fig. 1a. These are not present in any other reported system and no other study has applied such a combination to investigate the phenomenon of X-reactivation. This allowed us, for example, to retrieve for the first time allele-specific Hi-C maps of the transitory states of the X-reactivation process, that have not been described by any other laboratory, because other reprogramming systems are asynchronous and simply do not produce enough cells to perform Hi-C. PaX was instrumental in uncovering the links between 3D structural changes, chromatin opening and their impact on transcription on the X chromosome during reprogramming and X-reactivation, something that no other study has achieved so far.

3. Please explain why the authors chose D5 P-RFP+ which is closer to NPC at the RNA-seq or ATAC-seq level, as the intermediate, rather than D6 P-RFP+ or D6 X-GFP- which had obvious changes in RNA-seq. Is it because the positive rate of P-RFP cells is higher on the D5?

Author's Response: Activation of transcription is the ultimate consequence of the X-reactivation process. As our goal was to elucidate its potential causes, we decided that we should use a time point immediately before the onset of transcription. It was moreover due to technical reason, since isolating sufficient numbers of cells to perform Hi-C on a pure population of cells at a defined state prior to transcriptional X-reactivation proved to be more feasible at D5, as the vast majority of P-RFP+ cells on day 5 were X-GFP-. In contrast on day 6, cells had started to upregulate X-GFP and represented several subpopulations, each of which would have been difficult to isolate in sufficient numbers for high-resolution Hi-C analysis. This would have made cells from D6 for us less useful in order to gain information on the X-reactivation process.

This has been clarified in the main text:

Page 9: *We performed in situ Hi-C (Stadhouders et al. 2018; Rao et al. 2014) on NPCs, D5 P-RFP+/X-GFP- pre-iPSCs, and ESCs (Fig. 2a and Supplementary Fig. 2a), i.e., at the endpoints and immediately before the reactivation of gene expression.*

Page 33: *When we analyzed the Xi domain structure at day 5 of reprogramming, one day before the onset of full transcriptional reactivation, we found that more than 50% of TADs already displayed a significant increase in TAD connectivity (Fig. 6 and Supplementary Fig. 6).*

4. It is better to do biological experiment to confirm the relationship between Jpx and Xist.

Author's response: We absolutely agree with the reviewer's point of view, which is in line with the comment of reviewer 2 above, and therefore **have added new experiments directly showing the relationship of Jpx and Xist already in NPCs, by performing LNA-mediated knockdowns (Fig. 5h)**. This data shows that knockdown of Jpx RNA is sufficient for the downregulation of Xist RNA, while Jpx RNA is not affected by Xist RNA knockdown. This causal dependency of Xist RNA on Jpx expression and the strikingly similar kinetics of Xist and Jpx RNA downregulation during reprogramming (Fig. 5g) strongly support our hypothesis that Jpx downregulation might be an important driver of Xist downregulation during X-reactivation. These exciting new results strongly improve our manuscript and we thank the reviewer for suggesting this experiment.

5. There is no information about the Hi-C data in the manuscript. A supplementary of Hi-C experiment datasets with valid interaction read number should be provided.

Author's Response: **We have added a supplementary table with the valid interactions and we have expanded the description of the Hi-C data in the methods section.**

Minor concerns

1. There are wrong labeled in some figures. Such as "Fig 1C X-GFP~" should be revised as X-GFP-.

Author's Response: GFP~ has been replaced by GFPint. to avoid confusion.

2. The format of some references in the manuscript does not meet the requirements of this journal.

Author's Response: We have revised the references to meet the requirements.

REFERENCES

- Darrow, Emily M., Miriam H. Huntley, Olga Dudchenko, Elena K. Stamenova, Neva C. Durand, Zhuo Sun, Su-Chen Huang, et al. 2016. "Deletion of DXZ4 on the Human Inactive X Chromosome Alters Higher-Order Genome Architecture." *Proceedings of the National Academy of Sciences of the United States of America* 113 (31): E4504–12.
- Deng, Xinxian, Wenxiu Ma, Vijay Ramani, Andrew Hill, Fan Yang, Ferhat Ay, Joel B. Berletch, et al. 2015. "Bipartite Structure of the Inactive Mouse X Chromosome." *Genome Biology* 16 (August): 152.
- Froberg, John E., Stefan F. Pinter, Andrea J. Kriz, Teddy Jégu, and Jeannie T. Lee. 2018. "Megadomains and Superloops Form Dynamically but Are Dispensable for X-Chromosome Inactivation and Gene Escape." *Nature Communications* 9 (1): 5004.
- Fudenberg, Geoffrey, and Maxim Imakaev. 2017. "FISH-Ing for Captured Contacts: Towards Reconciling FISH and 3C." *Nature Methods* 14 (7): 673–78.
- Giorgetti, Luca, and Edith Heard. 2016. "Closing the Loop: 3C versus DNA FISH." *Genome Biology* 17 (1): 215.
- Giorgetti, Luca, Bryan R. Lajoie, Ava C. Carter, Mikael Attia, Ye Zhan, Jin Xu, Chong Jian Chen, et al. 2016. "Structural Organization of the Inactive X Chromosome in the Mouse." *Nature* 535 (7613): 575–79.
- Janiszewski, Adrian, Irene Talon, Joel Chappell, Samuel Collombet, Juan Song, Natalie De Geest, San Kit To, et al. 2019. "Dynamic Reversal of Random X-Chromosome Inactivation during iPSC Reprogramming." *Genome Research* 29 (10): 1659–72.
- Knaupp, Anja S., Sam Buckberry, Jahnvi Pflueger, Sue Mei Lim, Ethan Ford, Michael R. Larcombe, Fernando J. Rossello, et al. 2017. "Transient and Permanent Reconfiguration of Chromatin and Transcription Factor Occupancy Drive Reprogramming." *Cell Stem Cell* 21 (6): 834–45.e6.
- Krijger, Peter Hugo Lodewijk, Bruno Di Stefano, Elzo de Wit, Francesco Limone, Chris van Oevelen, Wouter de Laat, and Thomas Graf. 2016. "Cell-of-Origin-Specific 3D Genome Structure Acquired during Somatic Cell Reprogramming." *Cell Stem Cell* 18 (5): 597–610.
- Lee, J. T., and N. Lu. 1999. "Targeted Mutagenesis of Tsix Leads to Nonrandom X Inactivation." *Cell* 99 (1): 47–57.
- Minajigi, Anand, John Froberg, Chunyao Wei, Hongjae Sunwoo, Barry Kesner, David Cognigni, Derek Lessing, et al. 2015. "Chromosomes. A Comprehensive Xist Interactome Reveals Cohesin Repulsion and an RNA-Directed Chromosome Conformation." *Science* 349 (6245). <https://doi.org/10.1126/science.aab2276>.
- Polo, Jose M., Endre Anderssen, Ryan M. Walsh, Benjamin A. Schwarz, Christian M. Nefzger, Sue Mei Lim, Marti Borkent, et al. 2012. "A Molecular Roadmap of Reprogramming Somatic Cells into iPS Cells." *Cell* 151 (7): 1617–32.
- Rao, Suhas S. P., Miriam H. Huntley, Neva C. Durand, Elena K. Stamenova, Ivan D. Bochkov, James T. Robinson, Adrian L. Sanborn, et al. 2014. "A 3D Map of the Human Genome at Kilobase Resolution Reveals Principles of Chromatin Looping." *Cell* 159 (7): 1665–80.
- Schwarz, Benjamin A., Murat Cetinbas, Kendell Clement, Ryan M. Walsh, Sihem Cheloufi, Hongcang Gu, Jan Langkabel, et al. 2018. "Prospective Isolation of Poised iPSC Intermediates Reveals Principles of Cellular Reprogramming." *Cell Stem Cell* 23 (2): 289–305.e5.
- Stadhouders, Ralph, Enrique Vidal, François Serra, Bruno Di Stefano, François Le Dily, Javier Quilez, Antonio Gomez, et al. 2018. "Transcription Factors Orchestrate Dynamic Interplay between Genome Topology and Gene Regulation during Cell Reprogramming." *Nature Genetics* 50 (2): 238–49.
- Wang, Chen-Yu, David Cognigni, Hongjae Sunwoo, Danni Wang, and Jeannie T. Lee. 2019. "PRC1 Collaborates with SMCHD1 to Fold the X-Chromosome and Spread Xist RNA between Chromosome Compartments." *Nature Communications* 10 (1): 2950.

Wang, Chen-Yu, Teddy Jégu, Hsueh-Ping Chu, Hyun Jung Oh, and Jeannie T. Lee. 2018. "SMCHD1 Merges Chromosome Compartments and Assists Formation of Super-Structures on the Inactive X." *Cell*. <https://doi.org/10.1016/j.cell.2018.05.007>.

REVIEWERS' COMMENTS

Reviewer #1 (Remarks to the Author):

We thank the authors for their modifications. The revised version addresses all our concerns, except one point:

1. In response to our comment Nr. 12 the authors perform in silico mixing of Hi-C data between NPC and ESC. This is an interesting approach that could indeed address the question whether megadomains and TADs might co-exist within some cells. However, the authors only analyze, whether any mixing ratio exists that could explain relative domain scores of all domain categories. This analysis does not address the question whether megadomains and TADs co-exist, but rather analyses whether the different TAD categories appear with the same dynamics. Instead it would be useful to test whether a single mixing ratio exists that can reproduce the “strength” of the megadomain border and the domain score (of at least a subgroup of TADs) observed at day 5.

Reviewer #2 (Remarks to the Author):

I thank the authors for their explanations and additional experiments. The extra data and methods on the XX state of the cells and the functional data with Jpx knockdown are very useful.

I can understand that the requested DNA FISH is for the future, given the pandemic. I look forward to reading about this data in the future.

I would recommend one final wording addition. For the explanation of why ESC were chosen over iPSC, and why these are a valid choice, I think it would be helpful to include a sentence on this in the results. This would aid interpretation by the readers.

Reviewer #3 (Remarks to the Author):

No comments.

REVIEWERS' COMMENTS

Reviewer #1 (Remarks to the Author):

We thank the authors for their modifications. The revised version addresses all our concerns, except one point:

1. In response to our comment Nr. 12 the authors perform *in silico* mixing of Hi-C data between NPC and ESC. This is an interesting approach that could indeed address the question whether megadomains and TADs might co-exist within some cells. However, the authors only analyze, whether any mixing ratio exists that could explain relative domain scores of all domain categories. This analysis does not address the question whether megadomains and TADs co-exist, but rather analyses whether the different TAD categories appear with the same dynamics. Instead it would be useful to test whether a single mixing ratio exists that can reproduce the “strength” of the megadomain border and the domain score (of at least a subgroup of TADs) observed at day 5.

Author's response: We thank reviewer #1 for the comments and have addressed this in the text as follows.

“Our observation that the strength of the mega-domain boundary remained stable at D5 (Fig. 2d), suggested that the vast majority of cells still harboured a mega-domain, assuming similar mega-domain strength in cells as done previously¹⁹.”

Reviewer #2 (Remarks to the Author):

I thank the authors for their explanations and additional experiments. The extra data and methods on the XX state of the cells and the functional data with Jpx knockdown are very useful.

I can understand that the requested DNA FISH is for the future, given the pandemic. I look forward to reading about this data in the future.

I would recommend one final wording addition. For the explanation of why ESC were chosen over iPSC, and why these are a valid choice, I think it would be helpful to include a sentence on this in the results. This would aid interpretation by the readers.

Author's response: We thank reviewer #2 for the comments and have addressed this in the text as follows.

“We performed *in situ* Hi-C^{14,34} on NPCs to assess the inactive X, on D5 P-RFP+/X-GFP- pre-iPSCs to investigate possible structural changes immediately before the reactivation of gene expression, and on ESCs to capture cells with two active X and as an endpoint, as it has been shown previously that 3D genomes of iPSCs and of ESCs are overall highly similar⁵⁴ (Fig. 2a and Supplementary Fig. 2a).”

Reviewer #3 (Remarks to the Author):

No comments.